# Membrane-wide screening identifies potential tissue-specific determinants of SARS-CoV-2 tropism

**Ravi K. Dinesh**[1☯], **Chengkun Wang**[1,2☯¤], **Yuanhao Qu**[1,3☯], **Arjun Rustagi**[4,5☯],
**Henry Cousins**[1,6‡], **James Zengel**[7‡], **Xiaotong Wang**[1], **Trisha R. Barnard**[4],
**William A. Johnson**[1], **Guangxue Xu**[1], **Tianyi Zhang**[7], **Nicholas Magazine**[8], **Aimee Beck**[4],
**Lucas Miecho Heilbroner**[1], **Grace Peters-Schulze**[1], **Aaron J. Wilk**[4,6], **Mengdi Wang**[9],
**Weishan Huang**[8,10], **Brooke E. Howitt**[1], **Jan Carette**[6], **Russ Altman**[2,11]*,
**Catherine A. Blish**[4,6,12]*, **Le Cong**[1,2,3]*

1 Department of Pathology, Stanford University School of Medicine, Stanford, California, United States of America, 2 Department of Genetics, Stanford University School of Medicine, Stanford, California, United States of America, 3 Cancer Biology Program, Stanford University School of Medicine, Stanford, California, United States of America, 4 Division of Infectious Diseases and Geographic Medicine, Department of Medicine, Stanford University School of Medicine, Stanford, California, United States of America, 5 Department of Medicine, University of California, San Francisco (UCSF), San Francisco, California, United States of America, 6 Medical Scientist Training Program, Stanford University School of Medicine, Stanford, California, United States of America, 7 Department of Microbiology and Immunology, Stanford University School of Medicine, Stanford, California, United States of America, 8 Department of Pathology, School of Veterinary Medicine, Louisiana State University, Baton Rouge, Louisiana, United States of America, 9 Department of Electrical and Computer Engineering, Center for Statistics and Machine Learning, Princeton University, Princeton, New Jersey, United States of America, 10 Department of Microbiology and Immunology, College of Veterinary Medicine, Cornell University, Ithaca, New York, New York, United States of America, 11 Department of Bioengineering, Stanford University, Stanford, California, United States of America, 12 Chan Zuckerberg Biohub, San Francisco, California, United States of America

☯ These authors contributed equally to this work.
¤ present address: Department of Physiology, School of Basic Medical Sciences, Nanjing Medical University, Nanjing, China
‡ HC and JZ also contributed equally to this work.
* russ.altman@stanford.edu (RA); cblish@stanford.edu (CAB); congle@stanford.edu (LC)

## Abstract

While SARS-CoV-2 primarily infects the respiratory tract, clinical evidence indicates that cells from diverse cell types and organs are also susceptible to infection. Using the CRISPR activation (CRISPRa) approach, we systematically targeted human membrane proteins in cells with and without overexpression of ACE2, thus identifying unrecognized host factors that may facilitate viral entry. Validation experiments with replication-competent SARS-CoV-2 confirmed the role of newly identified host factors, particularly the endo-lysosomal protease legumain (LGMN) and the potassium channel KCNA6, upon exogenous overexpression. In orthogonal experiments, we show that disruption of endogenous LGMN or KCNA6 decreases viral infection and that inhibitors of candidate factors can reduce viral entry. Additionally, using clinical data, we find possible associations between expression of either LGMN or KCNA6

**Data availability statement:** All relevant data are in the manuscript and its Supporting Information files.

**Funding:** This work was supported by: National Institutes of Health grants R35HG011316 (L.C.), R01GM141627 (L.C.), K08 AI163369 (A.R.); the Baxter Foundation Faculty Scholar Award (L.C.); The Weintz Family COVID-19 Research Fund (L.C.); Bill and Melinda Gates Foundation Grant OPP1113682 (C.A.B.); Stanford University School of Medicine Dean's Postdoctoral Fellowship (R.K.D) and Pandemic Preparedness Hub Fellowship (T.R.B), and American Cancer Society Postdoctoral Fellowship (R.K.D.). The funders had no role in study design, data collection and analysis, decision to publish, or preparation of the manuscript.

**Competing interests:** The authors have declared that no competing interests exist.

and SARS-CoV-2 infection in human tissues. Our results identify potentially druggable host factors involved in SARS-CoV-2 entry, and demonstrate the utility of focused, membrane-wide CRISPRa screens in uncovering tissue-specific entry factors of emerging pathogens.

## Author summary

The SARS-CoV-2 virus is the cause of the COVID-19 pandemic which has resulted in the deaths of over 7 million people worldwide. SARS-CoV-2 enters cells by binding to the host protein ACE2 on the cell surface, but other proteins with varying distributions across the body can help enhance its entry. To identify new factors, we took advantage of a tool called CRISPR activation (CRISPRa) that can be used to express genes from across the body in a single cell type. Using CRISPRa we were able screen all known genes found on human membranes for their ability to alter SARS-CoV-2 infection. We uncovered a wide variety of genes but focused our studies specifically on two—the potassium channel KCNA6 and protease Legumain. We provide evidence for their roles by both using chemical inhibitors and genetic manipulation in cells where the genes are normally expressed and demonstrating these changes decrease infection. We further show for LGMN that altering its enzymatic activity or cellular location using mutations impedes its infection promoting effects. Our study identifies a role for novel factors in SARS-CoV-2 infection and suggest new avenues for further research.

## Introduction

The emergence of SARS-CoV-2 has led to the COVID-19 pandemic with over 770 million reported cases and over seven million reported deaths (WHO). Coronaviruses are a family of enveloped positive-stranded RNA viruses that cause respiratory and intestinal infections in birds and mammals [1]. Among the known human coronaviruses, four (229E, HKU1, NL63, and OC43) are widely circulating and cause mild infections, while three (SARS-CoV-1, Middle Eastern Respiratory Syndrome CoV, MERS-CoV, and SARS-CoV-2) are highly pathogenic [1].

SARS-CoV-2 enters cells in three major steps. The virus Spike protein first binds to its canonical receptor, angiotensin converting enzyme 2 (ACE2). This is followed by proteolytic processing of the Spike which can be carried out by several proteases, with TMPRSS2 and Furin being the most well-known. These steps lead to membrane fusion and consequent release of viral RNA into the host cell [2]. Recent studies have also implicated the binding of Spike protein to heparan sulfate [3] and cholesterol [4–6] as well as soluble-ACE2-mediated host cell attachment [7] as factors in viral entry, suggesting additional mechanisms that might be responsible for SARS-CoV-2 tropism and COVID-19 pathology.

Several research groups performed CRISPR knock-out loss-of-function (LOF) screens to find factors necessary for SARS-CoV-2 entry and replication [4,5,8–11]. A large number of hits were only found in one or a few screens but not in others [12]. Further, several experimentally confirmed entry factors, most notably neuropilins [13,14], were not reported as hits in any of the screens [12]. These discrepancies are likely due to the nature of LOF screens, which can only detect effects of expressed genes in the limited number of cell lines used, as well as the noise and confounding factors from genome-wide CRISPR screens. Recently, a study of host factors involved in rhinovirus infection demonstrated that surfaceome CRISPR knockout screens outperformed genome-wide screening in identifying true positives and eliminating false negatives [15]. While this study didn't account for potential non-surface bound membrane proteins that may be critical for viral entry, the results suggested targeted screening would likely improve the discovery of novel host factors for SARS-CoV-2.

Antibody-based therapies that target the SARS-CoV-2 Spike protein to inhibit ACE2 binding and prevent viral entry have been used to treat COVID-19 [16]. Similarly, current COVID-19 vaccines are highly effective in preventing symptomatic illness and function by triggering an immune response against the Spike protein [17]. However, newer viral strains with mutated Spike proteins have rendered both antibody therapies and vaccination potentially less effective [18–22]. Small-molecule compounds that target viral proteases are effective treatments for COVID-19, but these compounds act on later stages of the viral life cycle and have been largely unsuccessful in preventing illness in the specific case of SARS-CoV-2 infection [23]. Additionally, a number of groups have demonstrated that targeting the SARS-CoV-2 fusion process with peptide inhibitors has shown promise *in vitro* and in pre-clinical models, but proof of success in clinical trials remains outstanding [24–27].

Taken together, this suggests a critical need to gain further insight into SARS-CoV-2 entry mechanisms and develop therapeutics targeting virus entry. We turned to CRISPR activation (CRISPRa) screening of membrane proteins in cells with or without overexpression of the canonical ACE2 host receptor. Our CRISPRa screen identified potential novel host factors for viral entry, with reported expression in neuronal/sensory, respiratory, cardiovascular, and immune tissues. We validated two of the most interesting candidate genes–LGMN and KCNA6—using a combination of cDNA overexpression, mutagenesis, endogenous knockout, and pharmaceutical inhibition with pseudoviral and replication-competent SARS-CoV-2 infection assays. We further show that both LGMN and KCNA6 are dependent on the canonical factor ACE2, but their effects are not mediated through altering expression of ACE2 or other known SARS-CoV-2 co-factors. Importantly, both factors can promote infection in cellular contexts with almost undetectable levels of ACE2. Using previously reported scRNA-seq data from COVID-19 patients' bronchoalveolar lavage fluid [28], we show a strong positive relationship between LGMN expression and SARS-CoV-2 RNA levels. In summary, our work demonstrates the advantages of using membrane-focused functional genomics to identify host factors of SARS-CoV-2, offers potential targets for drug development, and suggests a useful approach to deciphering the infectious biology of emerging pathogens.

## Results

### CRISPRa screens identify putative host factors of SARS-CoV-2 susceptibility across tissues

To discover novel host factors involved in SARS-CoV-2 viral entry using CRISPRa screening, we first engineered HEK293FT cells with the synergistic activation (SAM) system (S1A Fig) and created lines with (ACE2 OE HEK293FT) or without (WT HEK293FT) ACE2 overexpression (S1B Fig). We then confirmed that these cells were capable of sgRNA-directed gene activation (S1D Fig) and ACE2-dependent infection using a SARS-CoV-2 D614G-mutant-Spike with a 19 amino acid C-terminal deletion (hereafter, D614G-Spike) pseudotyped lentivirus that confers zeocin resistance and GFP expression (S1C Fig). Finally, we generated a customized library of gRNAs targeting all known and putative membrane proteins (~6,213 genes totaling ~24,000 sgRNAs with non-targeting controls). Next-generation sequencing analysis demonstrated high levels of representation and minimal skewing, evidenced by a perfect guide match rate of 95.7% and a modest top-to-bottom skew ratio of 2.42. (S2A and S2B Fig).

To complete the screening workflow (Fig 1A) we transduced ACE2 OE and WT 293FT cells with the membrane-focused gRNA library, infected them with Sh-Ble expressing lentiviruses pseudotyped with D614G-Spike

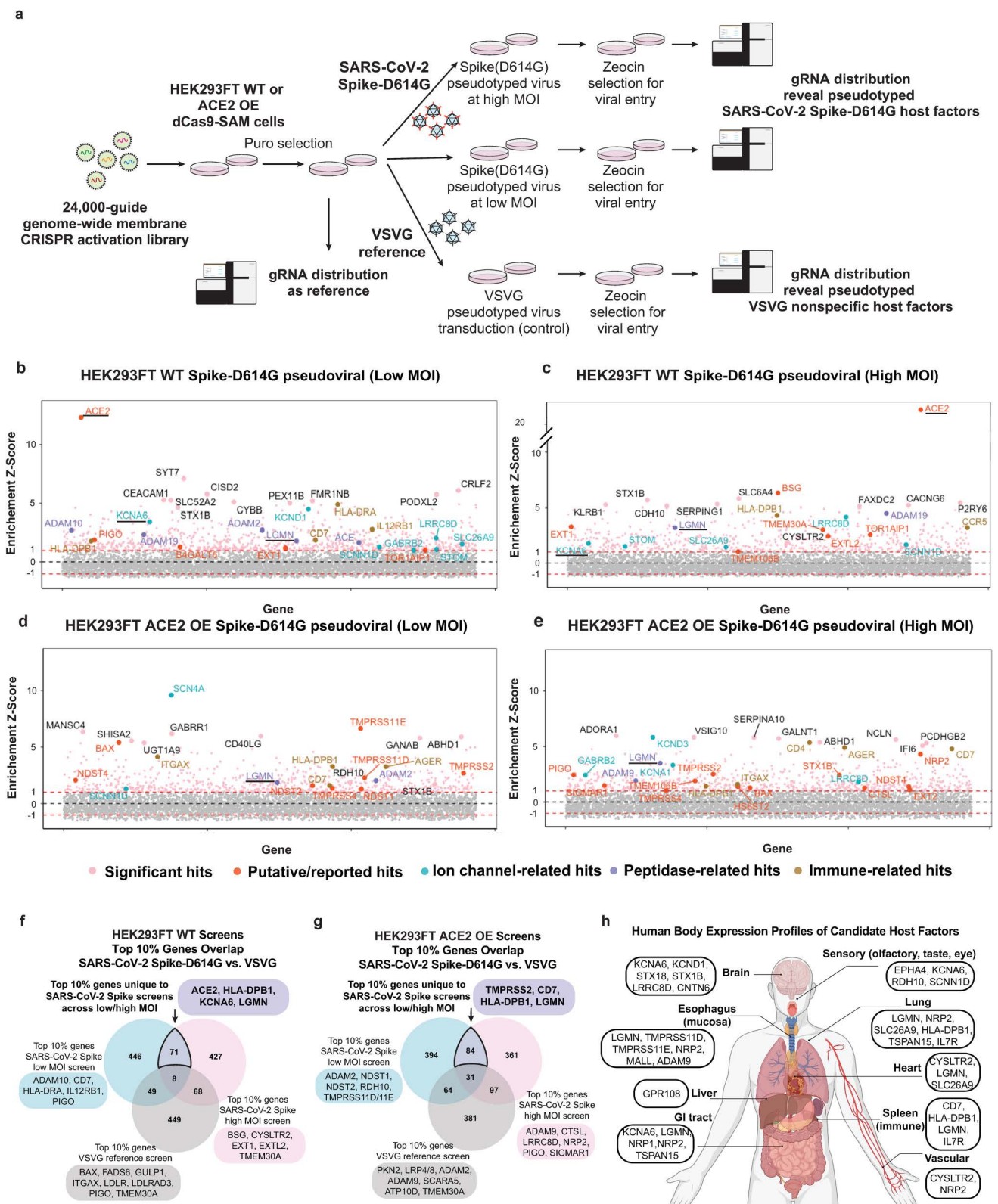

**Fig 1. Membrane-focused CRISPRa screening identifies potential host factors involved in Spike-dependent SARS-CoV-2 virus entry. a,** Screen pipeline showing different conditions used and the downstream analyses and validation workflow. **b-e,** Enrichment scores of CRISPRa screens across

different conditions with top hits highlighted and colored by their functional categories. Known (ACE2, TMPRSS2) and top-ranked new host factors are underlined. The enrichment cut-off score (generated using MAGeCK pipeline) was 1.0, and noted by a dashed line. **f-g,** Differential analysis of top 10% of hits from SARS-CoV-2 Spike and reference VSVG screens. The unique hits in SARS-CoV-2 screens help to identify putative virus-specific host factors from the screen. **h,** Tissue expression [29] body map to visualize top-ranking genes from the analysis of candidate host factors in panels **f-g**. Panels a,h were created with Biorender.

(SARS-CoV-2 group) or vesicular stomatitis virus envelope G protein (VSV-G group, as a reference), and selected with zeocin. Screens were completed across four conditions: WT 293FT or ACE2 OE at either high or low multiplicity of infection (MOI). The results demonstrated that our approach could identify factors known to promote D614G-Spike-dependent viral entry (e.g., ACE2, TMPRSS2, Neuropilin) as well as new genes and pathways (e.g., ion channels, immune genes, and proteases) expressed across tissues known to be susceptible to SARS-CoV-2 infection (Fig 1B–G). First, genes in the WT 293FT groups identified potential factors involved in ACE2-independent SARS-CoV-2 viral entry (Fig 1F). These included known genes such as the heparan sulfate synthesis enzymes EXT1 and EXTL2 [3] as well as unknown factors, such as potassium ion channel KCNA6 and the lysosomal/endosomal protease LGMN (Fig 1B,C and 1F). Similarly, the ACE2OE group identified a number of genes previously described to have roles in ACE2-dependent cellular entry, including members of the TMPRSS family and NDST enzymes (Fig 1D,E and 1G). New potential viral entry factors included NRP2, ion channels, and the immune genes CD7 and HLA-DPB1 (Fig 1G).

Critically, our control VSVG screens allowed us to rule out genes that were not specific to SARS-CoV-2 such as those involved in apoptosis and growth and pan-viral or lentiviral factors (e.g., ITGAX and the two subunits of the P4-ATPase Flippase Complex, ATP10D and TMEM30A) (Fig 1F–G). We also compared our CRISPRa screens with six recently published CRISPR LOF screens that utilized infection with live SARS-CoV-2 virus and cytopathic effect (CPE) as a readout. We first extracted the relative rankings of 4923 shared membrane proteins and generated a curated gene list of 18 validated virus entry factors. We then examined the ability of each CRISPR screen to identify these factors and found that 12 were among the top 10% of the highest ranked genes from our screens (S3 Fig).

Using the Genotype-Tissue Expression (GTEx) dataset [30], we analyzed expression of top-ranked genes across 24 tissues. We found that several hits, including STOM, LGMN, and TSPAN15, were found broadly across tissues (S4 Fig) while others showed specific expression in neuronal, cardiovascular, liver, and gastrointestinal tissues, offering potential insight into the mechanisms of SARS-CoV-2 viral entry in these cells [31–34] (Fig 1H). Notably, a significant number of our highly ranked genes are ion channels, transporters or receptors expressed in the brain and sensory systems, including, EPHA4, KCNA6, LRRC8D, and RDH10 (Fig 1H). Our top hits also included several genes highly expressed in immune cells (Fig 1H), such as CD7 and MHC-II components. Furthermore, Gene ontology and pathway enrichment analyses of screen hits identified several biological processes known to be involved in SARS-CoV-2 infection, including glycosylation, heparan sulfate synthesis, and peptidase/protease activities [3,34] (S5A and S5B Fig). While false positives are common in CRISPRa screens, these analyses suggest possible roles for putative host genes in respiratory, neuronal, cardiovascular, liver, gastrointestinal and immune cells.

Taken together, initial analysis of our screens offered potential explanations for the presence of SARS-CoV-2 in diverse tissues reported to be susceptible to SARS-CoV-2 [28,30,31] and suggested potential candidate factors for further study.

### Arrayed pseudoviral and replication-competent SARS-CoV-2 infection validates putative viral entry factors

To directly connect target protein expression with viral entry promotion, we performed arrayed validation of cDNA overexpressing WT 293FT and ACE2 OE lines with Spike and VSVG pseudotyped lentiviruses (Fig 2A–B). We selected genes based on results of overlap analysis (Fig 1F–G), pathway analysis (S4 Fig), and expression in disease-relevant tissues,

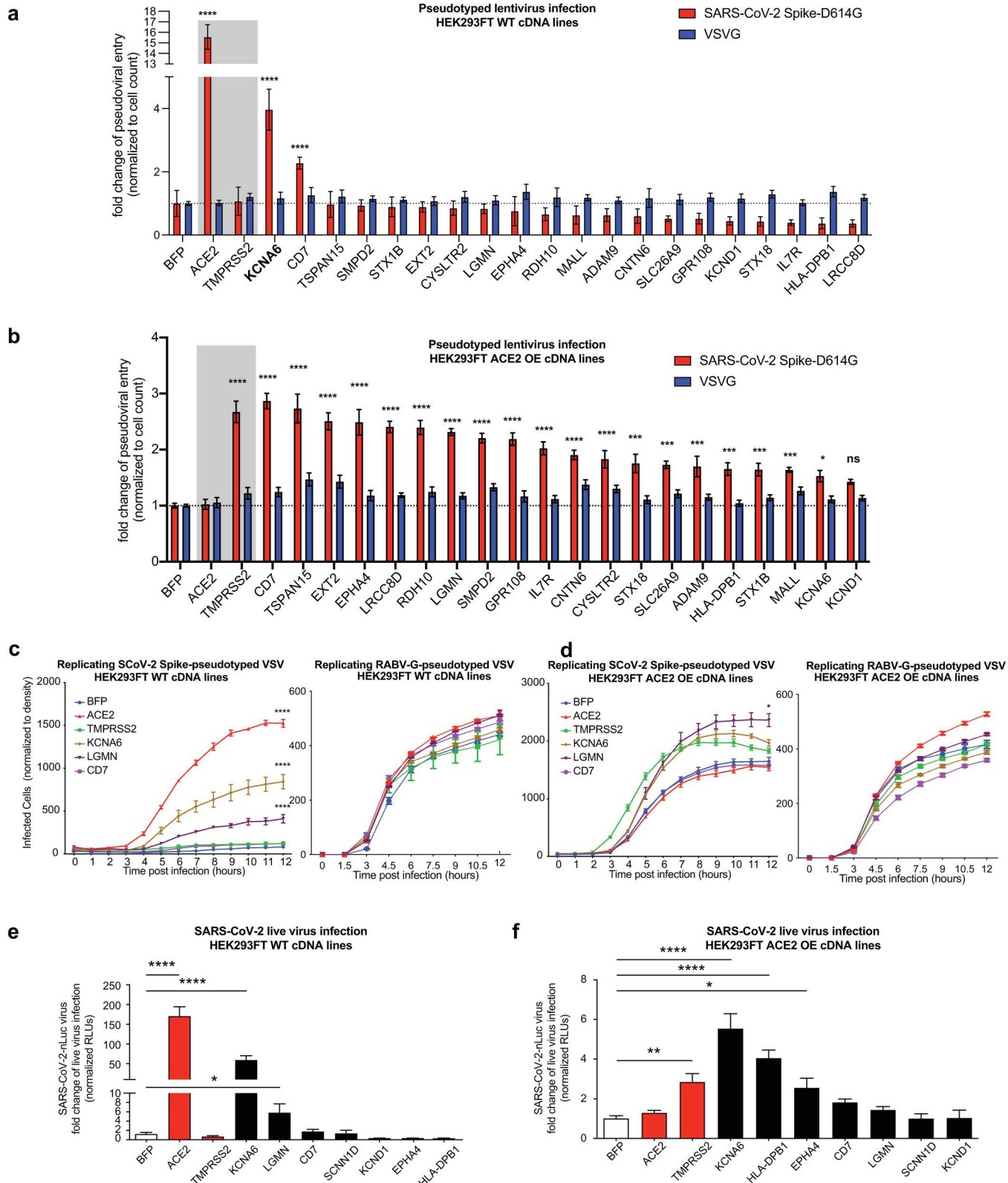

**Fig 2. Validation of top-ranked genes using pseudoviral and replication-competent SARS-CoV-2 infection assays.** a-b, Arrayed validation of top hits in cDNA overexpressing cell lines of individual genes using SARS-CoV-2 Spike-D614G pseudotyped lentiviral assay. The control VSVG-pseudotyped lentivirus results are shown side-by-side. Results are normed to the BFP control for each respective pseudotype. Data from two

independent experiments. **c-d,** Arrayed validation using time-lapse imaging of SARS-CoV-2 Spike-pseudotyped VSV infection in cDNA overexpression cell lines in 293FT WT cells **(c)** and 293FT ACE2OE (d). Companion control infection was performed using Rabies virus (RABV) G protein pseudo-typed VSV. Data from two independent experiments. **e-f,** Validation of top-ranked genes using SARS-CoV-2-nLuc virus infection. Data from three independent experiments. All data represent mean with SEM. Statistical analyses: for the panels **a-d**, using two-way ANOVA with correction for multiple comparisons during hypothesis testing, and for the panels **e-f**, one-way ANOVA with correction for multiple comparisons. *, $p < 0.05$; **, $p < 0.01$; ***, $p < 0.001$; ****, $p < 0.0001$.

focusing on the lung, respiratory tract, gastrointestinal tract, and immune cells (S5 Fig). Strikingly, in the WT 293FT condition, overexpression of KCNA6 promoted Spike-mediated pseudoviral infection, increasing viral entry ~4-fold above the control compared with ~15-fold for ACE2 cDNA (Fig 2A). In the ACE2-positive condition, overexpression of almost all cDNAs enhanced pseudovirus entry with CD7, EPHA4, LRCC8D, and LGMN overexpressing lines showing greater than 2-fold increases in Spike-mediated infection (Fig 2B). We also confirmed three of our hits: KCNA6, LGMN, and HLA-DPB1 promoted viral entry using lentiviruses pseudotyped with Spike proteins from the B.1.617.2 (Delta) and B.1.1.529 (Omicron) variants (S6 Fig). cDNA expression in cell lines that promoted Spike-dependent viral entry most robustly was verified using quantitative PCR (qPCR) and Western Blotting (S7A–B Fig).

In a complementary assay using replicating Spike-pseudotyped VSV with time-lapse imaging of virus infection, we found that two of the strongest hits validated by the pseudotyped lentiviral experiments, KCNA6 and LGMN, were able to similarly promote infection (Fig 2C–D). Parallel experiments with Rabies virus (RABV) G protein pseudotyped VSV in the same cell lines showed no cDNA-dependent effects (Fig 2C–D). The consistent results from two pseudoviral systems confirmed that effects from our genes of interest were specific to SARS-CoV-2 Spike, and not artifacts of the systems used.

Finally, we used a nanoluciferase-expressing SARS-CoV-2 reporter virus, icSARS-CoV-2nLuc [35], to measure if candidate overexpression could promote replication-competent virus infection. Consistent with our pseudovirus results, KCNA6 and LGMN significantly promoted SARS-CoV-2 infection compared with control groups in WT 293FT cells (Fig 2E–F). These results demonstrated that findings from our screen were translatable to replication-competent SARS-CoV-2 biology.

### Genetic or pharmaceutical inhibition of LGMN decreases SARS-CoV-2 viral entry

We selected the endo-lysosomal protease LGMN and the potassium channel KCNA6 for further analysis as both genes strongly promoted SARS-CoV-2 entry in our assays and have inhibitors that are either FDA-approved or under preclinical study [36,37]. We first sought to examine the roles of KCNA6 and LGMN in mediating viral entry in an endogenous context (Fig 3A).

LGMN, which encodes human legumain/delta-secretase, is a asparaginyl endopeptidase (AEP) with broad tissue distribution. Evidence suggests that its expression and activity increase with age [38], which is notable considering the association between COVID-19 morbidity and age. Our initial results (Fig 2E–F) indicated that LGMN overexpression could significantly increase replication competent SARS-CoV-2 infection of HEK293FT WT, but not ACE2OE cells despite showing significant differences in tests using pseudovirus (Fig 2B). The vector, pLX304 [39], used to express cDNAs in our initial validation work had the blasticidin resistance marker and GOI under the control of separate promoters which could lead to our cell lines staying resistant to selection despite silencing of GOI expression. We also noted that our LGMN expression vector had a suboptimal Kozak sequence flanking the LGMN coding sequence. As such, we hypothesized that extended culture might explain the weaker effect of LGMN expression in the context of ACE2 overexpression. Thus, we cloned the LGMN cDNA into a new expression vector with a Kozak consensus sequence and bicistronic expression under the control of an EF1a promoter using a 2A sequence (S8A Fig). Assays with live SARS-CoV-2 demonstrated that expression of LGMN in ACE2 OE cells using our new vector could significantly increase replication competent SARS-CoV-2 infection (S8B Fig).

As LGMN is expressed widely across tissues, we were interested to see if its ablation in a context of endogenous expression could lead to decreased SARS-CoV-2 infection. To that end, we transduced cell lines with endogenous LGMN

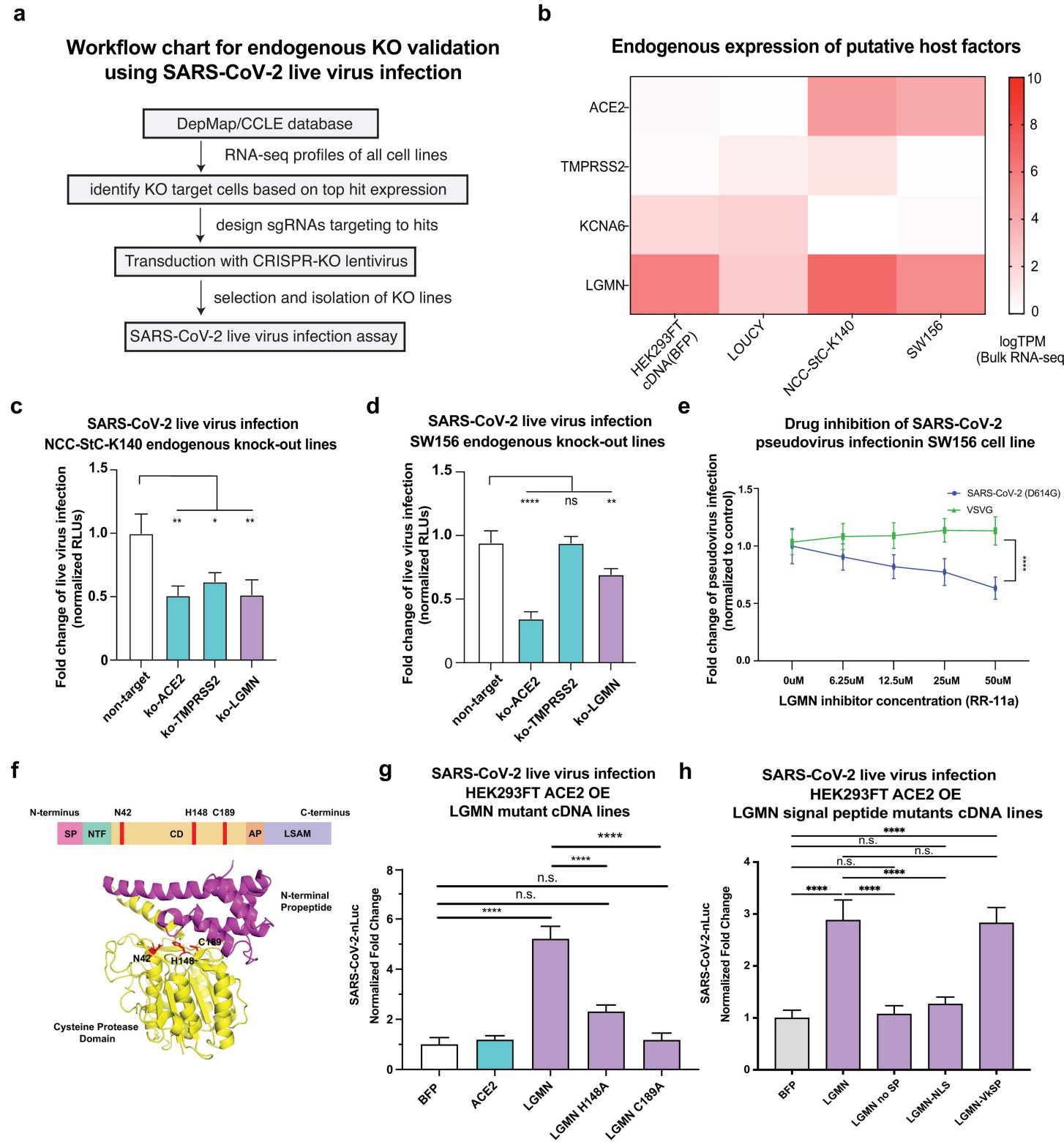

**Fig 3. Disruption of putative host factor LGMN decreases SARS-CoV-2 viral entry.** a, Workflow for endogenous knockout validation using SARS-CoV-2 virus infection. **b,** RNA-seq heat-map of gene expression in representative cell lines alongside cDNA-overexpressing reference lines. **c-d,**

SARS-CoV-2 live virus infection of NCC-Stc-K140 **(c)** and SW156 **(d)** cells perturbed with CRISPR-based loss-of-function constructs. Three guideRNAs were used per gene and results for the three independent lines were performed via individual arrayed infections and pooled for analysis. The entire infection was repeated twice to collect replicates for all cell lines. Results are normed to the BFP control/vehicle for each respective pseudotype. **e,** Data showing the dosage-dependent effect on infection of SARS-CoV-2 Spike-D614G or VSV-G pseudotyped lentivirus in LGMN-expressing SW156 cells treated with RR-11a, a specific inhibitor of legumain (LGMN). **f,** Structure and domains of human Legumain. SP: signal peptide, NTF: N-terminal fragment, CD: catalytic domain, AP: activation peptide, LSAM: legumain stabilization and activity modulation domain. **g-h,** SARS-CoV-2 live virus infection of ACE2OE 293FT cells expressing LGMN or LGMN catalytic **(g)** and signal peptide **(h)** mutant cDNA. Results are normed to BFP control. Data in **(g)** from three independent experiments and **(h)** from two independent experiments. All data represent mean with SEM. Statistical analyses: for **c-d,f** performed via one-way ANOVA with BFP or non-target as the control condition and for **e, g-h** with two-way ANOVA, all with correction for multiple comparisons during hypothesis testing. *, $p < 0.05$; **, $p < 0.01$; ***, $p < 0.001$; ****, $p < 0.0001$.

expression with lentiCRISPRv2 constructs encoding Cas9 plus gRNAs targeting either ACE2, TMPRSS2, or LGMN and subjected the knock-out cell lines to infection with replication competent SARS-CoV-2. We employed three guideRNAs targeting each gene where individual guideRNAs were measured independently (S9A–C Fig) and then pooled for plotting. First, we disrupted ACE2, TMPRSS2, and LGMN in the gastric carcinoma cell line NCC-Stk-K140, which expresses moderate levels of ACE2, low levels of TMPRSS2, and high levels of LGMN (Fig 3B) and serves as a potential model system for SARS-CoV-2 infection of the gut [40,41]. Disruption of any of these factors reduced infection with replication competent SARS-CoV-2 (Figs 3C and S9A) indicating that LGMN could promote infection in the context of tissues with ACE2 expression. We also disrupted the same three genes in the renal cell carcinoma line SW156, which expresses moderate levels of ACE2 and LGMN, but low to almost undetectable levels of TMPRSS2 (Fig 3B) and serves as a model for SARS-CoV-2 infection of the kidney [42,43]. Strikingly, we found that disruption of LGMN–but not TMPRSS2–lead to significant decreases in replication competent SARS-CoV-2 infection (Figs 3D and S9B), suggesting that LGMN acts independently of TMPRSS2 to promote infection.

## LGMN's catalytic activity and localization are critical for its infection promoting activity

To gain mechanistic insight into LGMN's role in SARS-CoV-2 entry, we tested the commercially available Legumain-targeting protease inhibitor RR-11a in LGMN expressing SW156 cells and found that increasing concentrations lead to a ~40% reduction in infection by Spike-pseudotyped lentiviruses at the highest concentration of drug, but not with VSVG pseudotyped viruses (Fig 3E). In order to further elucidate a role for LGMN's enzymatic activity in promoting Spike-dependent viral entry, we created LGMN cDNAs with mutations of critical catalytic residues [44] (Fig 3F) and expressed them in ACE2 OE cells. Live SARS-CoV-2 assays demonstrated that the H148A mutation ablated LGMN's SARS-CoV-2 promoting activity and the C189A mutation reduced viral infection to levels seen in the BFP control cells (Fig 3G).

To better understand the potential localization of LGMN with Spike protein, we imaged HEK293FT WT and ACE2OE cells with and without LGMN expression freshly (~4 hr) transduced with Spike-psuedotyped lentivirus (S10 Fig). The experiments showed that SARS-CoV-2 Spike protein co-localizes with LGMN and that this colocalization is more pronounced in the context of ACE2 overexpression (S10 Fig). In mammalian cells, Legumain localizes primarily to endosomes and lysosomes, but can also be found in the nucleus, cytoplasm, or cell surface of the cell [44–46]. To examine the role LGMN's cellular localization plays in viral infection we created LGMN signal peptide (SP) mutants. We first removed LGMN's signal peptide (LGMN no SP), which should lead to diffuse expression in the cytoplasm or extracellular secretion. We also created mutants in which the SP was replaced with either a nucleoplasmin nuclear localization signal (LGMN NLS) or an antibody kappa light chain signal peptide (LGMN Vk) to target LGMN to the cell surface. Expression of LGMN SP mutants in cells with ACE2 overexpression showed altered patterns of localization in line with alteration to LGMN's signal peptide sequence (S11 Fig). Infection with replication competent SARS-CoV-2 demonstrated that while removal of the signal peptide or substitution with an NLS ablated LGMN's infection promoting activity, targeting of LGMN to the cell surface (LGMN Vk) restored improved replication competent SARS-CoV-2 infection to WT levels (Fig 3H). In sum, our

experiments demonstrate that LGMN can facilitate SARS-CoV-2 viral entry in concert with ACE2 expression and its viral entry promoting activity is dependent on its catalytic activity and cellular localization.

## Disrupting the potassium channel KCNA6 decreases SARS-CoV-2 viral entry

KCNA6 belongs to a family of voltage gated potassium channels ($K_v$ family). To determine the specificity of KCNA6 in promoting SARS-CoV-2 entry, we overexpressed a panel of Kv family members in WT 293FT cells (Fig 4A). KCNA6 was the only tested family member whose exogenous expression led to significantly increased Spike-mediated pseudoviral entry (Fig 4A). Additionally, we found that KCNA6 maintained its ability to confer increased infection when using either B.1.351 (Beta) or B.1.617 (Delta) Spike-pseudotyped lentiviruses (Fig 4B–C).

We selected the T-cell acute lymphoblastic leukemia cell line Loucy as it expresses moderate levels of endogenous KCNA6 and LGMN, low levels of TMPRSS2, and low to undetectable levels of ACE2 (Fig 3B). We transduced the cells with lentiCRISPRv2 constructs targeting either ACE2, TMPRSS2, KCNA6, or LGMN and subjected the knock-out cell lines to infection with replication competent SARS-CoV-2. We found that loss of KCNA6 or LGMN significantly decreased viral infection by replication competent SARS-CoV-2 (Figs 4D and S9C). These results indicated that LGMN and KCNA6 may play a role in amplifying SARS-CoV-2 infection in the context of minimal/low expression of canonical SARS-CoV-2 factors.

Given the need for new pharmaceutical interventions that could alleviate SARS-CoV-2 spread and pathology, we next wanted to test the potential of an inhibitor of KCNA6 to prevent SARS-CoV-2 entry into tissues. 3,4-diaminopyridine (3,4-DAP), known commercially as amifampridine, is a potassium channel inhibitor that is FDA-approved for the treatment of Lambert-Eaton myasthenic syndrome (LEMS). Treatment of KCNA6 expressing Loucy cells with 3,4-DAP lead to a reduction, but not complete ablation, in infection by lentiviruses pseudotyped with D614G Spike (Fig 4E). Critically, this effect was Spike dependent as 3,4-DAP had no effect on infection by VSVG pseudotyped lentiviruses (Fig 4E).

Taken together, our data indicate that endogenous KCNA6 expression can facilitate SARS-CoV-2 infection with low/minimal presence of known SARS-CoV-2 entry factors, and that KCNA6 may be amenable to pharmaceutical targeting.

## LGMN and KCNA6 promote viral entry dependent on cellular ACE2 expression, but don't alter expression of known SARS-CoV-2 host factors

As previous reports have indicated that SARS-CoV-2 can infect wildtype 293T cells at low levels [47], we wanted to investigate the role of cellular ACE2 co-expression in KCNA6 and LGMN ability to promote SARS-CoV-2 infection. Using a monoclonal 293T cell line with ACE2 genetically disrupted (hereafter ACE2KO 293T), we found that overexpression of ACE2 could induce viral infection, but overexpression of either KCNA6 or LGMN in this context did not promote entry by SARS-CoV-2 live virus over background levels (Fig 5A). This contrasted with overexpression of cDNAs in the WT 293FT cells (Fig 2E) that had no detectable levels of ACE2 at the protein level (Fig 5B). This data indicates that KCNA6 and LGMN are dependent on cellular context of ACE2 expression for their infection promoting activity.

We then wanted to examine if KCNA6 and LGMN functioned by regulating the expression of ACE2 or other reported host factors for SARS-CoV-2 (S3 Fig). Western Blotting for ACE2 showed that exogenous expression of KCNA6 or LGMN did not increase levels of ACE2 (Fig 5B). Similarly, we performed qPCR (Figs 5C, S12A and S12B) and RNA-Seq (Figs 5D, S12C and S12D) on cells from the WT and ACE2 OE conditions and did not find significant differences in ACE2 or TMPRSS2 levels following KCNA6 or LGMN expression (Figs 5C, 5D, and S12, and S2 Data). Similarly, the expression of other reported host factors, such as Furin, CTSL and NRP1, was unchanged when measured by either RNA-Seq or qPCR following LGMN or KCNA6 overexpression in 293FT cells (Figs 5D, S12A and S12B, and S3 Data).

From these experiments, we conclude that cellular ACE2 co-expression is necessary for the effects of KCNA6 and LGMN in promoting SARS-CoV-2 viral entry. Our data indicate that KCNA6 and LGMN function in a context of cellular ACE2 expression to promote SARS-CoV-2 infection, and their effects are not mediated through altering expression other known host co-factors.

PLOS Pathogens

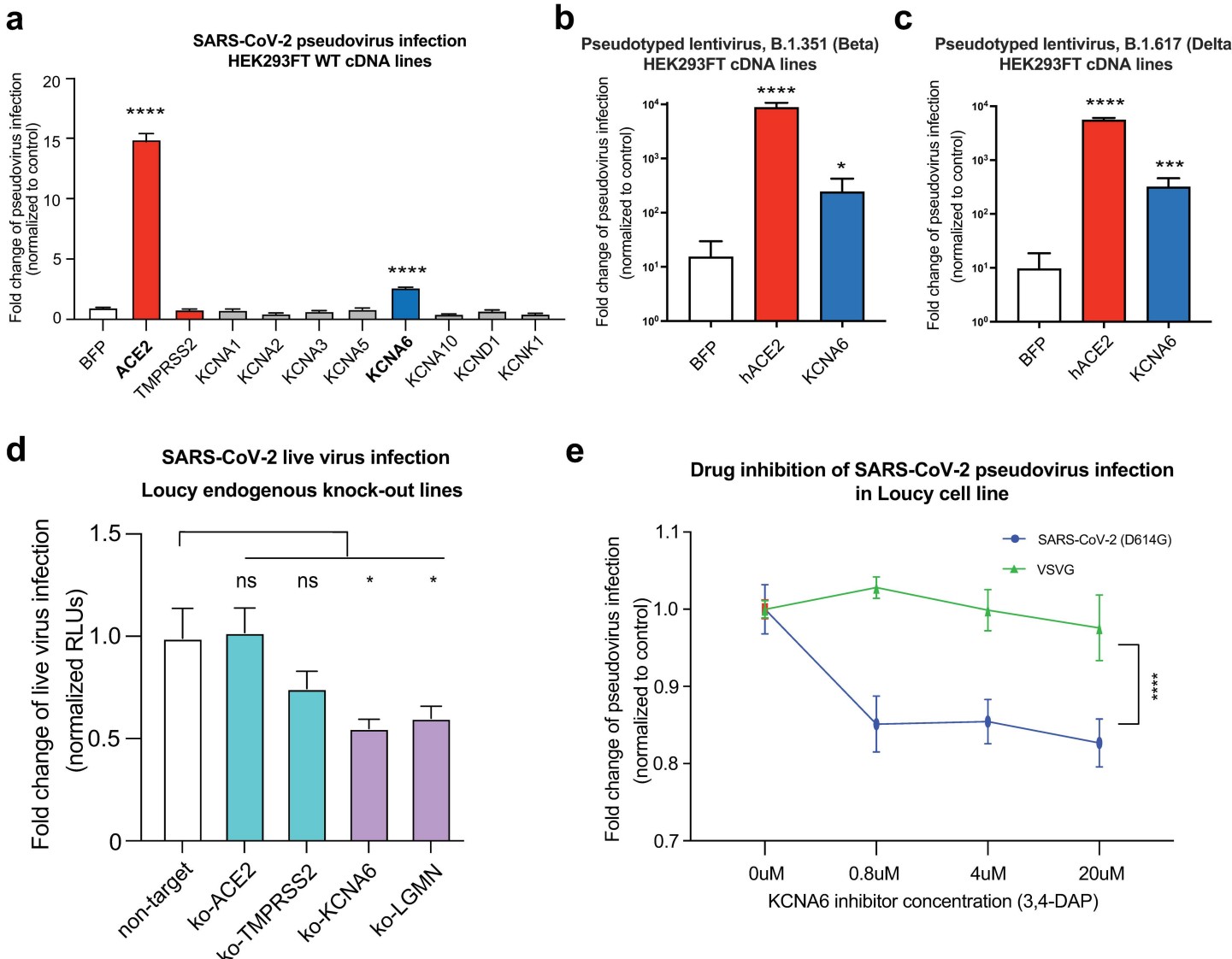

**Fig 4. KCNA6 is a putative SARS-CoV-2 host factor.** a, Infection of SARS-CoV-2 Spike-D614G pseudotyped lentivirus in 293FT cells overexpressing Kv channel members. Three independent experiments performed. **b-c,** Infection of cDNA overexpressing 293FT cells with lentivirus pseudotyped with either B.1.351 (Beta) or B.1.617 (Delta) Spike. n = 5. **d**, SARS-CoV-2 live virus infection of Loucy cells perturbed with CRISPR-based loss-of-function constructs. Three guideRNAs were used per gene and results for the three independent lines were performed via individual arrayed infections and pooled for analysis. The entire infection was repeated twice to collect replicates for all cell lines. **e,** Inhibition of SARS-CoV-2 viral entry with FDA-pproved compound 3,4-Diaminopyridine (Amifampridine), a potassium channel inhibitor. Data showing the dosage-dependent effect on infection of SARS-CoV-2 Spike-D614G or VSV-G pseudotyped lentivirus in KCNA6-expressing Loucy cells treated with 3,4-AP. Results are normed to the BFP control/vehicle for each respective pseudotype. Data from two independent experiments. All data represent mean with SEM. Statistical analyses: for **a-d** are performed via one-way ANOVA with BFP or non-target as the control condition and for **f** with two-way ANOVA, all with correction for multiple comparisons during hypothesis testing. *, $p < 0.05$; **, $p < 0.01$; ***, $p < 0.001$; ****, $p < 0.0001$.

## LGMN and KCNA6 are expressed at sites of SARS-CoV-2 infection and pathology

LGMN is widely expressed across tissues including in the lung and respiratory tract (Fig 6A). Examining previously published single-cell RNA-seq data from the bronchoalveolar lavage fluid (BALF) of COVID-19 patients [28] indicated that LGMN expression was pronounced in plasmacytoid dendritic cells and macrophages (Fig 6B) and had a positive

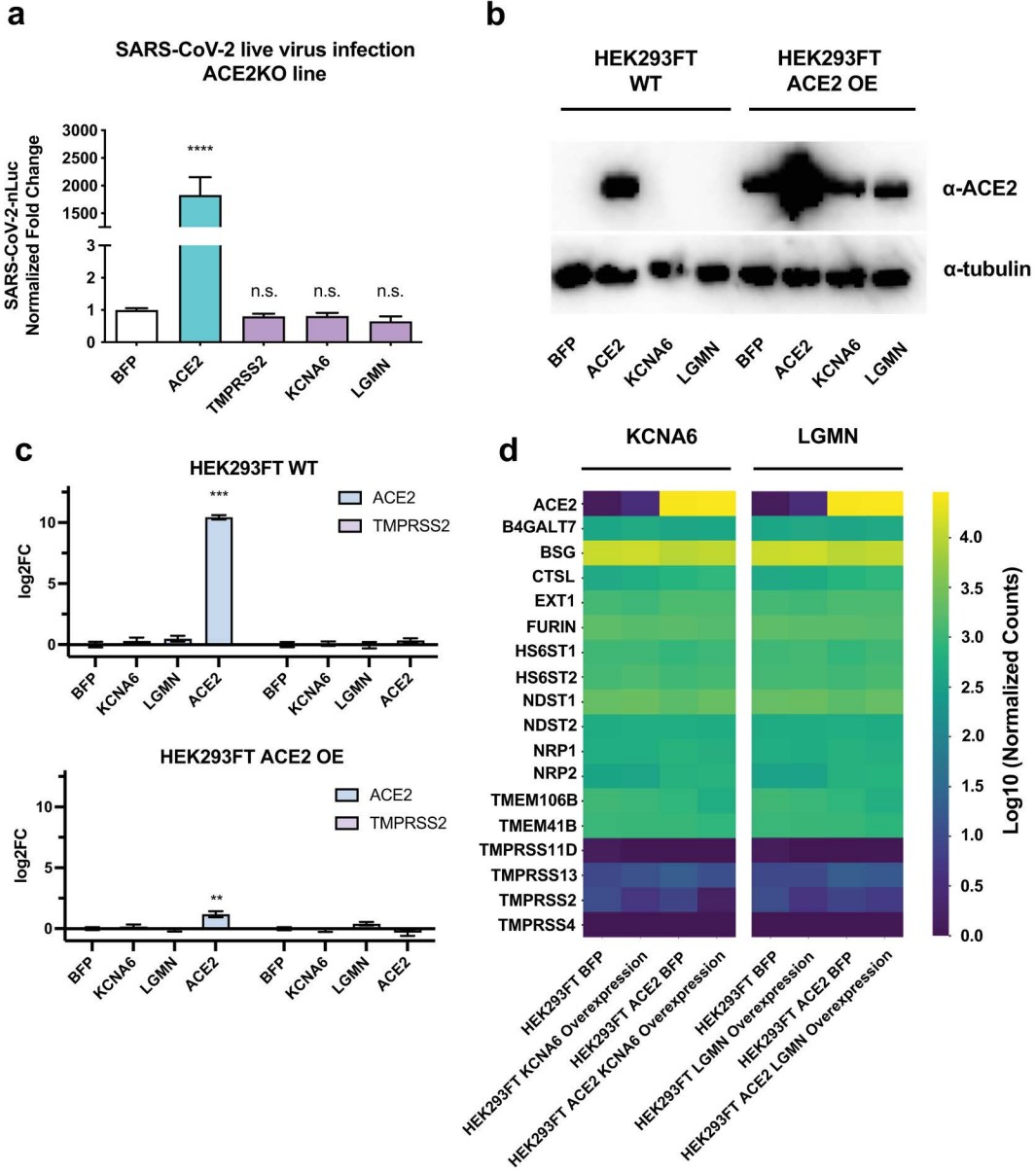

**Fig 5. KCNA6 and LGMN promote SARS-CoV-2 infection dependent on ACE2 but independent of other known SARS-CoV-2 host factors.**
a, Left, SARS-CoV-2 live virus infection of ACE2KO 293T cells overexpressing cDNAs. Data from two independent experiments. b, Western Blot of ACE2 levels in WT and ACE2 OE 293FT cells overexpressing cDNAs. c, Quantitative RT-PCR analysis of ACE2 and TMPRSS2 levels upon the over-expression of indicated cDNAs. d, Whole transcriptome analysis via RNA-seq of the KCNA6 and LGMN over-expressing cell lines, compared with BFP-expressing control lines, two independent replicates were sequenced. All data represent mean with SEM. Statistical analyses in a were performed via one-way ANOVA with BFP serving as the control condition, with correction for multiple comparisons during hypothesis testing, in c with two-tailed t test using Welch's correction with BFP as the control condition. *, $p < 0.05$; **, $p < 0.01$; ***, $p < 0.001$; ****, $p < 0.0001$.

correlation with levels of SARS-CoV-2 viral RNA (Fig 6C). Moreover, using data from patients with healthy, moderate, or severe disease [28] we observed that LGMN expression was positively correlated with COVID-19 disease severity (Fig 6D–E). These results are in line with other studies demonstrating SARS-CoV-2 infection of LGMN+ macrophages in the lungs of patients that subsequently contribute to disease progression [48]. Taken together with our *in vitro* data, this

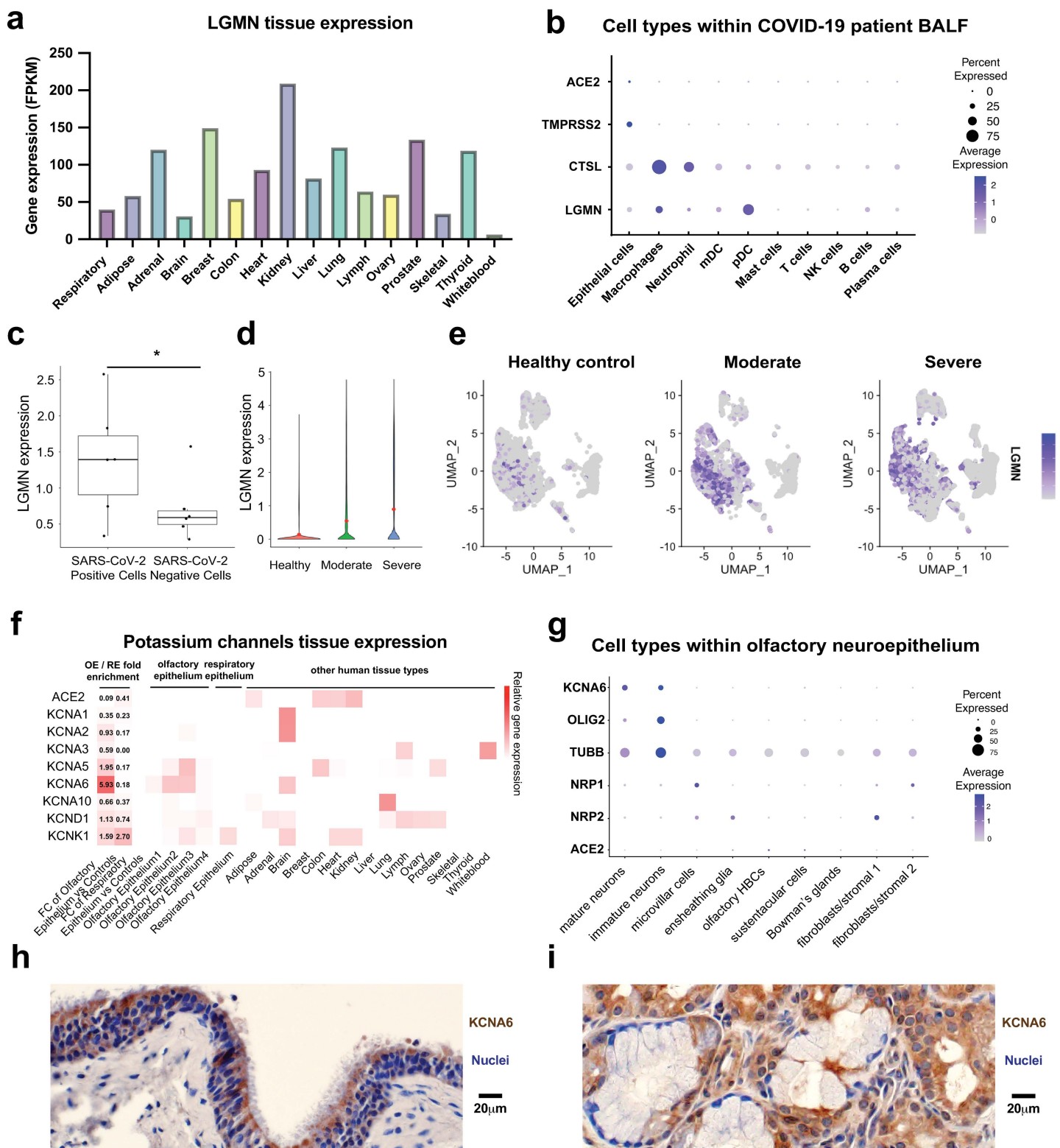

**Fig 6. KCNA6 and LGMN are expressed in disease-relevant cell types.** a, Expression of LGMN across human tissues [49,50]. **b**, Expression of ACE2, TMPRSS2 and LGMN in different cell types from BALF. **c,** Average expression levels of LGMN between SARS-CoV-2 positive and negative cells from severely affected patients. Each dot represents a severely affected patient. Two-sided Student's t-test. **d-e,** Healthy controls (n = 4); COVID-19

patients (moderate) n = 3; COVID-19 patients (severe), n = 6. **d,** LGMN expression from single cell RNA-seq data. **e,** UMAPs depicting the expression of LGMN. **f,** Expression of ACE2 and KCNA family genes across human tissues. Left two columns show the fold change (FC) of expression comparing olfactory epithelium (OE) or respiratory epithelium (RE) vs. control tissues. Columns on the right show the expression levels across individual samples. **g,** Expression of ACE2 and KCNA6 in the single-cell RNA-seq data of olfactory neuroepithelium [51]. Sizes of the dots indicate the proportion of cells having greater-than-zero expression while the color indicates mean expression. **h-i**, Immunohistochemistry analysis of KCNA6 protein expression in human sinonasal tissues using archival clinical samples.

analysis suggests that in addition to increasing host cell susceptibility, increased expression of LGMN over the course of infection could be a marker of poorer clinical outcomes. Further experiments will be needed to establish a causal connection between LGMN expression and susceptibility using *in vivo* models.

Similar to LGMN, KCNA6 overexpression or disruption led to significant alternations in SARS-CoV-2 infection (Figs 2C–F and 3F). Using two bulk RNA-Seq datasets [49,50], we found that KCNA6 is enriched in nasal tissues and the brain relative to other KCNA family members and is expressed in the olfactory epithelium, while the canonical SARS-CoV-2 receptor ACE2 is nearly absent (Fig 6F). We reanalyzed data from a recent single-cell RNA-seq (scRNA-seq) analysis of olfactory neuroepithelium in adult humans [51] using the Salmon-Alevin pipeline—which can handle multiple-mapped reads—and were able to readily detect high levels of KCNA6 expression with the standard GRCh37 reference (S13A–C Fig). Likewise, adjusting the genome reference to remove overlapping transcripts allowed us to detect high KCNA6 expression using the CellRanger pipeline (S13B Fig). Analysis of KCNA6 at the single cell level showed robust expression within human olfactory neuroepithelium (S13E Fig), with especially high expression in OLIG2 + oligodendrocytes (S13D–F Fig).

Despite low expression of ACE2 and other known host factors, olfactory epithelium in COVID-19 patients are reportedly infected [52–54] with two recent studies using COVID-19 autopsy samples demonstrating that SARS-CoV-2 is detected in OLIG2+ or TUBB(TUJ1)+ oligodendrocytes [14,55]. Our analysis indicates KCNA6 is likely present in this virus-infected cell type (Fig 6G). KCNA6 immunohistochemistry on archival clinical samples of human sinonasal tissues showed KCNA6-positive cells with epithelial or serous gland origin (Fig 6H-I) Our findings indicate the presence of KCNA6 could be relevant for sensory aspects of COVID-19 symptoms and suggest the potential for uncovering new ACE2-dependent entry mechanisms [56], but more stringent studies would be needed to further understand this effect.

## Analysis of real-world clinical data uncover potential druggable targets for COVID-19

To utilize our screening results to explore other potential pharmaceutical options, we systematically examined 254 of our top SARS-CoV-2 screen hits by constructing a drug-target interaction network based on data available from the Drug Bank [57] (S14 Fig). To evaluate which drugs could potentially alter SARS-CoV-2 viral entry either through direct interaction with SARS-CoV-2 Spike or modulation of known co-factor gene entry, we ranked the drug compounds by normalized degree centrality (NDC) and the proportion of drug interactors that were CRISPRa screen hits (degree ratio) (S15A Fig). We observed marked enrichment of several drug classes with a propensity for potassium-channel targeting, including antidepressant, anticonvulsant, and antipsychotic agents, many of which have support in the literature for a role in modulating SARS-CoV-2 viral entry [49,50]. To provide a clinical assessment of our findings, we performed a retrospective analysis of COVID-19 hospitalization from claims data of 7.8 million Medicare Advantage Part D members (S15B Fig and S1 Table). We identified 98 drugs, including broad-spectrum anticonvulsants and thiazide diuretics, whose odds ratio was significant (corrected p < 0.05) (S2 Table) as well as highly ranked in our screen drug-target network (p < 1e–11, Mann-Whitney U test; S15C Fig), suggesting an association between a compound's ability to modulate viral entry genes and its associated risk of COVID-19 hospitalization. Using propensity score matching (PSM) we evaluated whether common drugs targeting these ion channels were associated with decreased risk of hospitalization and found a significant risk-associated effect for loop diuretics, SSRI antidepressants, and broad-spectrum anticonvulsants that persisted after PSM, with hazard ratios >1.2 consistently and significantly (S15D Fig).

## Discussion

In the work presented here we demonstrate the utility of CRISPRa screening to provide insight into the tropism of an emerging pathogen. CRISPR LOF approaches are powerful, but limit discovery of host factors to genes expressed in the cell line used. A potential path to addressing these concerns involves doing screens in multiple cell lines representing a variety of cell types, but this is laborious and necessitates prior knowledge of viral tropism. Our CRISPRa screening approach overcomes these limitations by selecting cell lines with limited or no susceptibility and allowing for the unbiased determination of factors that promote viral entry. We show here that previously unknown factors expressed across a diverse set of tissues—such as LGMN, KCNA6, and HLA-DPB1—can stimulate SARS-CoV-2 viral entry upon exogenous expression.

Most strikingly, we show that the protease LGMN and the potassium channel KCNA6 can potentiate SARS-CoV-2 entry in exogenous and endogenous cellular contexts of high or low ACE2 expression (Figs 2E, 2F, 3C, 3D and 4D). These effects are dependent on ACE2, but not mediated through altering expression of other known host co-factors, suggesting mechanisms of entry deserving of further investigation (Fig 5A–D). We performed three arrayed gRNA knockout experiments so that the cell lines were independently tested and then pooled for data analysis, to minimize the possibility that the observed phenotypes were from off-target editing. However, future studies, ideally using monoclonal lines, will be needed to fully validate knockout effects. Additionally, we demonstrate in an endogenous context that pharmaceutical inhibition of either factor is capable of dampening viral entry. While our initial tests were positive, given potential interactions of target proteins with other factors, and possible secondary or off-target effects from chemical inhibitors, additional studies, particularly *in vivo* studies of virus infection, will be needed to further validate these findings. Nevertheless, our data suggest that LGMN and KCNA6 are potential actionable targets for future work.

LGMN, is expressed in macrophages and a recent large-scale scRNA-seq study of BALF cells from the lungs of healthy and cystic fibrosis subjects found that LGMN expression was specifically pronounced in interstitial macrophages (IMs) [58]. Recent work has shown that IMs are a predominant target of SARS-CoV-2 in the lung and a major driver of inflammation[51]. Other work demonstrating infection of lung macrophages in SARS-CoV-2 patients has found that CD163+LGMN+ exposed to SARS-COV-2 promote fibrosis and Acute Respiratory Distress Syndrome (ARDS) in COVID-19 [48]. These studies in tandem with our findings suggest that targeting LGMN could ameliorate COVID-19 associated inflammation in the lung.

Our data support that KCNA6 is expressed in nasal OLIG2+/TUJ1+oligodendrocytes (Fig 6), which have been shown to be a cell type susceptible to SARS-CoV-2 infection in the olfactory epithelium [14,55]. Olfactory and taste dysfunction are common and persistent symptoms of COVID-19 [56], and a small fraction of hospitalized patients suffer from serious neurological conditions, such as delirium, encephalopathy and stroke [43,56]. The degree to which these neurological effects are due to infection of neuronal cells or the side effects of an inflammatory state are still poorly understood [43,59]. Experimental studies have demonstrated that brain organoids are susceptible to SARS-CoV-2 infection despite the low levels of ACE2 receptor expression [59,60]. Such infection appears to be poorly correlated with levels of ACE2, TMPRSS2, or NRP1 expression [59]. This suggests that neuronal-specific cofactors, such as KCNA6, may work in synergy with the ACE2 receptor to promote viral entry. It is important to note that our data does not provide evidence that KCNA6 functions through a direct interaction with the Spike protein, and it remains to be tested whether KCNA6 expression promotes infection through protein-protein interactions or an indirect effect on cell physiology.

While our manuscript was under preparation, four groups published genome-wide CRISPR activation screens to identify host factors involved in SARS-CoV-2 biology [61–64]. Strikingly, our validated hits–KCNA6 and LGMN–were not identified in these CRISPRa screens indicating that our membrane-targeted screening approach focusing on the viral entry portion of the viral life cycle uncovered host factors that standard approaches missed. It should be noted that these recent CRISPRa screens, including ours, had significantly more noise compared with loss-of-function knockout screens. Thus, the variability of activation screening led to relaxed statistical cut-offs and should be considered when evaluating factors reported by CRISPRa studies including the current one.

Our study used SARS-CoV-2 Spike-pseudotyped lentiviruses in our initial CRISPRa screening. As such, our discovery approach does not consider the potential role of other SARS-CoV-2 components, such as the envelope (E) or nucleocapsid (N) proteins. Our focused membrane sgRNA library likely reduced screen noise—as host factors involved in viral entry are rarely non-membrane-associated—but may not detect potentially interesting proteins that act indirectly to boost infection.

Taken together, the studies presented here offer insight into SARS-CoV-2 viral tropism and yield potential new targets for drug development or drug repurposing. Our work also presents a platform that can be applied to future emerging pathogens to rapidly uncover factors involved in viral susceptibility across tissues.

## Materials and methods

### Plasmids and constructs

The following constructs were obtained from Addgene: lenti dCAS-VP64_Blast (61425, a kind gift from Feng Zhang), lenti-CRISPRv2 (52961, a kind gift from Feng Zhang), pLX304 (25890, a kind gift from David Root), pCMV-VSVG (8454, a kind gift from Bob Weinberg), pTwist EF1 Alpha nCoV-2019-Spike-2xStrep (141382, a kind gift from Nevan Krogan), pCI-VSVG (1733, a kind gift from Garry Nolan), pXPR_502 (96923, a kind gift from David Root and John Doench), VSV-eGFP-dG vector (31842, a kind gift from Connie Cepko), and pcDNA3.1-hACE2 (145033, a kind gift from Fang Li). EF1a-Hygro, EF1a-ACE2-2A-Hygro, and EF1a-EGFP-2A-ZeoR were Gibson cloned into FUGW using the aforementioned Addgene plasmids as PCR templates. Spike variants (Sd19, Sd19 D614G, Sd19 D614G-V5) were Gibson cloned into the pCI backbone by replacing the VSVG protein in pCI-VSVG. BFP, ACE2, LGMN, and LGMN mutant cDNAs were cloned into pLV EF1a-2A-BSD through using NEBuilder HiFi DNA Assembly Master Mix (NEB E2621). Amino acid sequences for SP mutants are as follows: WT (VWK-VAVFLSVALGIGA), NLS (KRPAATKKAGQAKKKK), Vk (ETDTLLLWVLLLWVPGSTGG). Host factor cDNA containing vectors were ordered from DNASU or Genecopoeia as either lentiviral transfer plasmids or gateway entry vectors. Gateway entry cDNAs were subsequently cloned into the destination vector pLX304 using the Gateway LR clonase kit (Invitrogen 11791019). A list of cDNA vectors is provided in S3 Data. gRNAs were cloned into LentiCRISPRv2 using the NEB Golden Gate Assembly Kit (NEB E1602L). A list of targeted genes and their corresponding gRNAs is provided in S4 Table.

### Cell line culture, generation, and validation

293FT, SW156, and A549 cells were maintained in DMEM with high glucose and Glutamax (Gibco 10566016) supplemented with 1% Pen-Strep (Gibco 15-140-122), 1% NEAA (Gibco 11140050) and 10% FBS (BenchMark) at 37°C, 5% $CO_2$. Loucy cells were maintained in Advanced RPMI1640 (Gibco 12633012) supplemented with 10% FBS (BenchMark) at 37°C, 5% CO2.

293FT/dCas9-VP64 cells were generated by transducing 293FT cells with dCAS-VP64_Blast lentivirus. Cells were selected with 10 ug/mL blasticidin and kept on the concentration of selection except in cases of double or triple selection, wherein the doses were reduced to 5 ug/mL. HEK293FT WT and HEK293FT ACE2 OE cell lines were generated by transduction of either a pLV-EF1a-hACE2-2A-Hygro lentivirus or a pLV-EF1a-Hygro lentivirus into either 293FT cells or 293FT/dCas9-VP64 cells. 293FT or 293FT/dCas9-VP64 cells were selected in 500 ug/mL hygromycin and the doses were reduced to 250 ug/mL in cases of double or triple selection. For cDNA overexpression studies, HEK293FT WT and HEK293FT ACE2 OE cells were transduced with cDNA overexpression lentiviruses and selected with blasticidin (10 ug/mL).

Genetic disruption of SW156, NCC-Stc-K140, and Loucy cells were carried out by transducing cells with LentiCRISPRv2 vectors carrying sgRNAs of interest. Cells were then selected in puromycin (SW156 - 0.5 ug/mL, NCC-Stc-K140 - 1.0 ug/mL, Loucy - 0.2 ug/mL). ACE2 KO 293T cells were purchased from Creative Biogene.

### Lentiviral, pseudoviral production and transduction

Pseudotyped lentiviruses were produced using PEI or JetOptimus transfection reagent (Polyplus) according to manufacturer's protocols with a ratio of 2:2:1(transfer plasmid: pCMV-dR8.91: Envelope plasmid). For VSVG pseudotyped

lentivirus, pCMV-VSVG was used as an envelope plasmid. For SARS-CoV-2 Spike protein pseudotyped lentivirus, Spike Sd19 (Spike protein with C-terminal deletion of 19 amino acids) and Sd19 D614G, as well as additional variant (Beta, Delta, Omicron) plasmids cloned in house (as described in Plasmids and Constructs) were used as an envelope plasmid. After 4 hours, the media was replaced and ViralBoost (Alstem, VB100) was added to a 1X concentration. 48 hours after transfection pseudotyped lentiviruses were collected, passed through a 0.45 um filter, and frozen down at -80°C.

For large scale production of the membrane protein sgRNA library or production of SARS-CoV-2 Spike protein or Spike variant viruses, 293FT cells in 10 cm dishes were transfected with JetOptimus at the aforementioned ratios, media was replaced 4 hours after transfection, and ViralBoost was added to 1X. At 48 hours, viral supernatant was collected and passed through a 0.45 um filter. For all VSVG psuedotyped viruses, viral supernatant was aliquoted immediately and stored at -80°C. For SARS-CoV-2 Spike and Spike variant pseudoviruses, Lentivirus Precipitation Solution (Alstem) was added to 1X. Viral precipitation was carried out according to the manufacturer's protocol. Following precipitation, lentiviruses were concentrated 10X using either DMEM or Ultraculture media (Lonza, discontinued) and frozen down at -80°C.

## Pseudoviral assay using pseudotyped lentivirus

Every batch of pLV-GLuc-2A-EGFP lentivirus was titered by the addition of $1 \times 10^4$ cells to wells of a 96 well plate followed by the addition of varying volumes of virus. Media was added to a final volume of 100 uL and polybrene was added to a final concentration of 8 ug/mL. Cells were spinfected at 1000 x g, 32°C for 45 min before being returned to the 37°C incubator. After 24 hours, supernatant was removed from the wells and replaced with 200 uL of fresh media. 72 hours after the spinfection, cells were washed with PBS and resuspended in FACS Buffer (1X PBS, 2% FBS), and assayed for percentages of cells that were GFP positive by flow cytometry. All conditions were done in duplicate. For the experimental assay, $1 \times 10^4$ control or modified cells were placed in the wells of the 96 well plate and a volume of virus that gives an MOI of approximately 0.08-0.1 (as determined by the aforementioned titering experiment) was added to each well before the addition of polybrene (8 ug/mL final concentration) and media to a final volume of 100 uL in each well. The assay was then completed as described above for the titering experiment.

## Inhibitor pseudoviral assay

Inhibitor assays for SW156 cells use 96-well plates coated with Poly-D-Lysine (Thermo Fisher, A3890401) at a concentration of 50 ug/mL for 2 hours at room temperature. The plates were then washed with PBS three times, and $1 \times 10^4$ cells were plated in a final volume of 100 uL of culture media with inhibitors. The next day, 6 uL of diluted SARS-CoV-2 Spike D614G or VSVG pseudotyped lentiviruses (for an ~ MOI of 0.05-0.15) with added polybrene were added to each well for a final concentration of 8 ug/mL polybrene. Plates were spinfected and assayed as described above. 3,4-Diaminopyridine (Sigma-Aldrich D7148, 3,4-DAP) was diluted in PBS from a stock concentration of 50mM. RR-11a (MedChemExpress, HY-112205A) was diluted in DMSO from a stock concentration of 100mM. The cells were pretreated with inhibitors at specific concentration for 24 hours, then infected with pseudovirus.

## Membrane protein CRISPR activation library

A list of all known membrane associated proteins was derived from Chong *et al*. [65], with the following adjustments and manual curation: we included representative olfactory receptor genes and removed pseudogenes or fusion genes that may introduce noise during screening. This refined list was used to pull 4 sgRNAs for each gene from the Calabrese human CRISPR activation pooled library. For genes not included in the Calabrese pool, guides were manually designed using the CRISPOR tool (http://crispor.tefor.net/). For all the final sgRNA sequences, we ensured the starting nucleotide was either G or A by adding a starting G when necessary to maintain efficient Pol-III transcription initiation. The full detail of the customized human membrane protein CRISPRa library design is in S1 and S2 Data. Oligonucleotide pools were synthesized by TwistBio (CRISPR activation pool) or IDT DNA (additional validation pool) with Esp3I/BsmBI recognition

site and PCR amplification sequences appended to the sgRNA sequence. The oligo sequence template was 5'-CATGTTGCCCTGAGGCACAGCGTCTCACACC [guide sequences, 20 or 21 nt] GTTTCAGTCTTCCGTCACATTGG CGCTCGAGA-3'. A set of primers (forward: CATGTTGCCCTGAGGCACAG and reverse: CCGTTAGGTCCCGAAAGGCT) was used to amplify the oligo pool using the manufacturer's protocol (Twist Bioscience, detailed in Twist oligo pool amplification guidelines). The PCR product was column purified using the Monarch PCR and DNA Cleanup Kit (NEB T1030S) and cloned into pXPR_502 via Golden Gate cloning using the Golden Gate Assembly kit BsmBI-v2 (NEB 1602L). The product was isopropanol precipitated, electroporated into Stbl4 electrocompetent cells (Invitrogen 11635018) with a Micropulser Electroporator (Bio-Rad) in 0.1 cm cuvettes. Cells were allowed to recover in 1 mL of recovery media at 30°C and then amplified in large scale at 25°C for 17 hours in 2-YT Broth. Plasmid DNA was prepped using QIAGEN Plasmid Plus Maxi Kit and sequenced to confirm library coverage and distribution.

### CRISPRa screening in 293FT cells

For each screen replicate 100 x 10$^6$ 293FT/dCas9-VP64 cells with or without ACE2 overexpression were transduced in total in 6 well plates. In each well, 3 x 10$^6$ cells were combined with a volume of CRISPRa membrane library viral supernatant to give an MOI of 0.3 before the addition of polybrene (Millipore TR-1003-G) to a working concentration of 8 ug/mL and add culture media to a final volume of 2 mL. Plates were spinfected in a tabletop centrifuge at 1000 x g for 45 min at 32°C. Following spinfection, 2 mL of culture media was added to each well and returned to 37°C incubators. 12 hours post-spinfection, viral supernatant was removed and cells from each well were split into individual 15 cm dishes at a final volume of 25 mL. 48 hours post-spinfection, puromycin (Gibco A1113803) was added to each plate of transduced and mock-infected cells at a final concentration of 0.5 ug/mL. Cells were selected until mock infected cells were completely killed and transduced cells had recovered to a 90% confluence with minimal cell death in the presence of puromycin selection.

sgRNA containing 293FT/dCas9-VP64 cells were maintained at a minimal cell number of 24 x 10$^6$ cells to prevent loss of representation. On the day of the spinfection, 24 x 10$^6$ library cells for each condition (ACE2-OE or WT) were harvested for gDNA. For each screen replicate 100 x 10$^6$ library cells for each condition were transduced in total in 6 well plates. For each well, cells were combined with viruses pseudotyped with either SARS-CoV-2 D614G Spike protein or VSVG envelope and carrying the EF1a-BleoR-2A-EGFP transfer vector at an approximate MOI or either 0.01 (low) or 0.1 (high). Media was added to a final volume of 2 mL with polybrene (Millipore TR-1003-G) at a working concentration of 8 ug/mL before being spinfected at 1000 x g for 45 min at 32°C. 2 mL of media was added to each well and the plates returned to the 37°C incubator until 12 hours post-spinfection when virus containing media was removed and each well was split into individual 15 cm dishes. Zeocin (Gibco R250-01) was added to transduced and mock-infected cells at a final concentration of 500 ug/mL. Cells were selected until mock infected cells were completely killed (approximately 5–7 days post Zeocin addition). Cells for each condition were then pooled and harvested for gDNA extraction.

### Genomic DNA isolation, guide RNA amplification and quantification

Genomic DNA of screening samples were extracted using Quick-DNA Midiprep Plus Kit (Zymo Research) following the manufacturer's protocol. Then, 5 ug of genomic DNA was added per 50 uL PCR reaction mixed staggered primers (synthesized by IDT DNA, Forward Primer: 5' AATGATACGGCGACCACCGAGATCTACACTCTTTCCCTACACGAC-GCTCTTCCGATCT[Stagger, 0–7nt]TTGTGGAAAGGACGAAACACC-3' Reverse Primer: 5'-CAAGCAGAAGACGGCA TACGAGAT [8nt-barcode]GTGACTGGAGTTCAGACGTGTGCTCTTCCGATCTTCTACTATTCTTTCCCCTGCACTGT-3') to increase the base diversity. At least 4 PCR reactions were used per sample to ensure adequate coverage. PCR reactions were then pooled and column purified using Monarch PCR and DNA Cleanup Kit (NEB T1030S), visualized on a gel, and column purified following gel extraction. The amplified products were quantified, and normalized by concentration, followed by sequencing using Illumina Miseq Reagent Kit v3 (150 cycles). Saturation analysis was done to confirm sequencing saturation.

## Computational analyses of CRISPR activation screens

The sequencing data was deconvoluted using bcl2fastq function (illumina). Reads per sgRNA were counted and processed using the standard Mageck pipeline with an output of RRA and gene ranks. We averaged the positive RRA scores of the biological replicates in each condition and calculated -log (average RRA) scores. Then, enrichment z-scores were calculated by z-normalizing -log (average RRA) scores in each condition. A cutoff of enrichment z-score of 1 was initially used to identify the hits.

GTEx v8 tissue specific enrichment was performed using the Multi Gene Query function available on the GTEx website [24,43]. Gene set overlap analysis was done using top 10% hits in each condition via GSEA-mysigDB (http://www.gsea-msigdb.org/gsea/msigdb/annotate.jsp) with Gene Ontology for Molecular Function. Functional interaction networks were constructed using Reactome FIPlugin in Cytoscape (http://apps.cytoscape.org/apps/reactomefiplugin).

## Comparative analyses of SARS-CoV-2 loss of function screens

SARS-CoV-2 loss of function screens data were directly obtained from the supplementary materials of Wei *et al*. [5], Daniloski *et al*. [9], Hoffmann *et al*. [10], Wang *et al*. [4], Zhu *et al*. [63]. and Baggen *et al*. [8]. on March 31st 2021. Relative rankings of 4923 membrane genes were extracted from each screen. For screens with multiple conditions, the best ranking across the conditions was used. A hit is defined as top 10% if the ranking is equal or smaller than 492.

## Generation of focused validation libraries

For WT HEK293T and ACE2-OE HEK293T screens, we separately selected the top hits that showed up in multiple biological replicates and MOI conditions and curated lists of genes for the focused pooled validation: 523 genes for WT library and 542 genes for ACE2-OE library. We added 2 guides per gene on top of the original library (6 guides per gene total) and 30 non-targeting guide RNAs as control. The oligonucleotide pools were also synthesized by Twist Bioscience. The plasmid library and lentivirus were made as described in the previous section.

## Cellular RNA extraction, cDNA generation, and quantitative RT-PCR

The cellular total RNA of validation cell line samples were extracted using Direct-zol RNA Miniprep Kits (Zymo Research) following the manufacturer's protocol. Then, 20 ng of total RNA was used in one-step quantitative RT-PCR reaction using *Power* SYBRGreen RNA-to-CT *1-Step* Kit (Thermo Fisher). All data are the product of at least two biologically independent experiments and six technical replicate experiments performed per experimental condition. qRT-PCR reactions were run on a Bio-Rad CFX384 Touch Real-Time PCR Detection System (Bio-rad) following the manufacturer's protocol. Multiple unpaired t tests with Welch's correction were used for all qRT-PCR analysis in current study.

## Total RNA-seq from cell lines and whole transcriptome analysis

For RNA-seq analysis of the validation cell lines, the total RNA from cDNA over-expressing cell line samples were extracted using Direct-zol RNA Miniprep Kits (Zymo Research) following the manufacturer's protocol. At least 5ug of each total RNA samples were prepared and subject to total RNA-Seq processing via reverse transcription and cDNA second strand synthesis (Azenta/Genewiz). At least 25 million reads were sequenced for each of the RNA-seq samples with at least 2 replicates per sample for all RNA-seq experiments. The sequencing was performed using HiSeq/NovaSeq (Illumina) and then raw reads were aligned using STAR, followed by differential expression analysis using the DESeq2 package (DESeq v1.44.0) following the recommended workflow, with a Log2-Fold Change threshold of 0.3219 (log-base 2 of 1.25) and a FDR-corrected p-value threshold of 0.05. Downstream visualization was performed in R (R v4.4.0) using the EnhancedVolcano (EnhancedVolcano v1.22.0) and ggplot2 (ggplot2 v3.5.1) in custom scripts.

## Analysis of RNA-seq and scRNA-seq data

The human olfactory epithelium RNA-seq data were directly obtained from Olender *et al.* [49] (https://www.ncbi.nlm.nih.gov/pmc/articles/PMC4982115/) and visualized using pheatmap in R. For scRNA-seq of the olfactory epithelium, patient 2 and patient 3 data (BAM file) from Durante *et al.* [51]. were downloaded from NCBI SRA portal (GSE139522). Because of the overlap of the references, we built a customized version of the GRCh37 reference by deleting the references of RP11-234B24.4 and GALNT8 in GRCh37.87. Then the data were aligned using the customized GRCh37 reference via CellRanger 6.0 (10X Genomics). For Alevin single-cell analysis, standard genome references (GRCh38 and GRCh37) built with Salmon (https://github.com/COMBINE-lab/salmon) were used to process the same datasets as CellRanger. The Salmon - Alevin pipeline was used with default parameters and adjustment of the 10x kit version according to the data-set specifications, namely 10x-v3 for patient2 and 10x-v2 for patient3. The resulting data matrices were processed using Seurat following the methods mentioned in Durante *et al*. [51] (https://github.com/satijalab/seurat). Bronchoalveolar lavage fluid (BALF) data from Liao *et al.* [28] was acquired from https://github.com/zhangzlab/covid_balf and then processed and visualized using Seurat in R.

## Flow cytometry analysis and surface marker staining assays

For ACE2 staining, cells were washed with 1X PBS and then incubated with biotinylated SARS-CoV-2 Receptor Binding Protein (RBD) (ACRO Biosystems SPD-C82E9) at a concentration of 4 ug/mL in FACS Buffer (1X PBS with 2% FBS) for 30 min at room temperature. The cells were washed twice with FACS Buffer and then incubated with streptavidin-Alexa 488 (Thermo Fisher, S11223) at a concentration of 2 ug/mL for 30 min at room temperature. Cells were washed twice with FACS Buffer and analysis was carried out on a Cytoflex Flow Cytometer.

## Western blotting

Cells were pelleted, washed with 1X PBS, and lysed in RIPA buffer (Cell Signaling Technology 98306). Immunoblotting was performed with the following primary antibodies: V5 (Thermo R960-25) and KCNA6 (Sigma HPA021516). All procedures and dilutions were performed according to the manufacturer's recommended protocol.

## Immunohistochemistry

All tissues were obtained from the Stanford Tissue Bank in compliance with all institutional guidelines. The tissue blocks were sectioned and then the immunohistochemistry was conducted on series of 4-μm-thick formalin-fixed, paraffin-embedded tissue sections using rabbit anti-KCNA6 polyclonal antibody (Sigma HPA014418), applying the Vectastain Elite ABC Universal Plus Kit according to their standard protocol. The following antibodies and dilutions were used: KCNA6 polyclonal antibody (HPA014418, Millipore-Sigma), and the dilution factor for all KCNA6 stainings is 1:500. The antigen retrieval for each stain included a 20-minute heat-activated epitope retrieval in pH 6 citrate buffer using a steamer.

## Replicating vesicular stomatitis virus (VSV) pseudovirus generation

Recombinant VSV expressing eGFP in the 1st position (VSVdG-GFP-CoV2-S) was generated as previously described [4]. The plasmid to rescue this virus was generated by inserting a codon optimized SARS-CoV2-S based on the Wuhan-Hu-1 isolate (Genbank:MN908947.3), which was mutated to remove a putative ER retention domain (K1269A and H1271A) into a VSV-eGFP-dG vector (Addgene, Plasmid #31842) in frame with the deleted VSV-G. The control virus VSVdG-RABV-G SAD-B19 was also generated by inserting Rabies virus G in the same vector. Both viruses were rescued in 293FT/VeroE6 cell co-culture and amplified in VeroE6 cells and titrated in VeroE6 cells overexpressing TMPRSS2. Sequencing of the amplified virus revealed an early C-terminal Stop signal (1274STOP) and a partial mutation at A372T (~50%) in the ecto-domain. Similar adaptive mutations were found in a previously published VSVdG-CoV2-S [66].

### Pseudovirus infection assay using replicating VSV pseudovirus

HEK293FT cells were plated in clear 96-well plates at $2x10^4$ cells per well approximately 24 hours prior to infection in 100 uL of media containing 10% FBS. Cells were infected with VSVdG-CoV2-S or VSVdG-RABV-G at an MOI of 0.1. Infection was performed by diluting virus in media without FBS and adding 150 uL of diluted virus per well. After addition of virus, the plate was spun at 900 x g for 60 minutes at 30°C. Infection was tracked over time using an Incucyte system (Sartorius) in a 37°C and 5% $CO_2$ incubator using 4x magnification and detecting GFP. GFP+ cells were counted using Incucyte Analysis software and data was reported as GFP positive foci per well after normalization to confluence.

### Replication-competent SARS-CoV2 live virus infection assay

ic-SARS-CoV-2-nLuc [35] was obtained from BEI Resources (NR-54003). The virus was passaged twice in VeroE6-TMPRSS2 cells, filtered on Amicon 100kD filters (Millipore Sigma), washed three times in sterile PBS, resuspended in DMEM with 2% FBS, aliquoted, and subsequently titered by plaque assay on VeroE6-TMPRSS2 cells using a 1.2% Avicel (FMC Biopolymer) overlay. All nLuc virus assays were performed in 96 well white bottom plates (Cell Star 655083). For SW156 and 293FT cells, cells were assayed on poly-D-lysine (Thermo Fisher, A3890401) coated plates prepared as described above. Cells were added to plates 16 hours prior to infection. After determining the optimal MOI for each cell line, cells were then infected for 2 hours at MOI 0.1 (for 293FT) or 0.5 (for SW156) or 1 (for Loucy), washed, and incubated for 48 hours before assessment by lytic Nano-Glo assay (Promega) and read on a GloMax plate reader (Promega). Infections and plate reading occurred inside class II biosafety cabinets under biosafety level 3 (BSL-3) conditions. All experiments using viruses were approved by the Administrative Panel on Biosafety (APB) at Stanford University.

### Immunofluorescence staining and confocal imaging

293FT WT or ACE2 overexpression cells with or without exogenous LGMN expression were seeded at $5x10^4$ cells per well in a 24-well plate approximately 24 hours prior to the transduction with Spike-V5tag pseudotyped lentivirus. Cells were collected at 4 hrs after transduction. 293FT cells with LGMN or LGMN signal peptide variants were seeded at $5x10^4$ cells per well in a 24-well plate before collection. Collected cells were fixed with 4% paraformaldehyde solution (Santa Cruz Biotechnology) for 15 minutes, then permeabilized with 0.1% Triton X-100 (Sigma-Aldrich) for 10 minutes at room temperature. Cells were washed with PBS and incubated for 1 hour at room temperature in blocking buffer (4% FBS in PBS), followed by primary antibody staining overnight at 4°C with mouse anti-V5 antibody (GenScript) and/or rabbit anti-LGMN antibody (Novus). After removal of Primary antibodies, the cells were washed with PBS 3 times, then incubated with Alexa Fluor 594 Donkey anti-Mouse IgG (Jackson ImmunoResearch) and/or Alexa Fluor 488 Goat Anti-Rabbit IgG (Jackson ImmunoResearch) secondary antibodies for 1.5 hours at room temperature. Cells were washed with PBS and stained with DAPI (Thermo Fisher) for 5 minutes, then mounted in Fluoroshield mounting medium (Sigma-Aldrich).The stained cells were imaged on the Stellaris confocal microscope platform (Leica), and images were analyzed with ImageJ.

### Drug-target network analysis

Genes were labeled as screen hits by first selecting the top quartile of genes by enrichment score, then removing genes that were also hits in the VSVG screen (>95th percentile by enrichment score). A list of drug-gene interactions was obtained from DrugBank and used to generate a bipartite graph with two node classes, drugs and genes, and edges representing known interactions between protein-coding genes and FDA-approved drugs. From this graph, the "full network," a subgraph was generated containing only genes that were screen hits and their associated drug interactions (the "screen-hits network"). Drugs were labeled according to DrugBank classifications and were ranked by normalized degree centrality as defined below:

$$ndc = n^{screen\ hits}/m^{screen\ hits}$$

where $n^{screen\ hits}$ is the node degree in the screen-hits network and $m$ is its maximum theoretically possible degree in the same network.

Drug nodes were separately ranked by "degree fraction," as defined below:

$$degree\ fraction = n^{screen\ hits}/n^{full\ network}$$

where $n^{screen\ hits}$ is the node degree in the screen-hits-only network and $n^{screen\ hits}$ is the node degree in the full network.

When calculating aggregate rankings for drug classes, we included only classes with greater than 10 members in the full network. Statistical tests are Mann-Whitney U tests and Spearman rank-order correlation as appropriate, unless stated otherwise.

## Claims data

We obtained de-identified Medicare Advantage Part D (MAPD) administrative claims data for patients from a large US health insurance provider. The dataset contained medical and pharmaceutical claims codified as International Classification of Diseases, Tenth Revision, Clinical Modification (ICD10-CM) and National Drug Codes (NDC) identifiers.

## Database-wide drug screen

For the initial claims database screen, drugs were defined by their generic names and used by individuals represented in claims data between July 1, 2019, and January 31, 2020. We selected the screening study cohort to be all the COVID-19 related hospitalized members and 1:10 exactly matched non-hospitalized members based on: age (+/- 1), gender, race, socioeconomic status (SES) index (+/- 0.5), living in counties from New York, New Jersey and Connecticut or counties outside the New York, New Jersey and Connecticut tri-state area, and diagnosis of diabetes without chronic complications, congestive heart failure, chronic pulmonary disease, myocardial infarction, metastatic cancer, liver disease, renal failure, peptic ulcer disease, and hypertension. We included unique drugs with over 2,000 users in the 1:10 exact matched cohort in the screening study. For each drug, we considered individuals drug-exposed if they had any reimbursed prescription claims during the study period. We considered individuals non-drug exposed if they had zero reimbursed prescription claims for the analyzed drug during the study period. We calculated the log odds of COVID-19 hospitalization as a binary outcome, comparing drug-exposed to non-drug-exposed individuals, across drug claims meeting our minimum sample size threshold. Odds ratio significance levels were adjusted for multiple hypothesis testing using a Benjamini-Hochberg correction.

## Construction of cohorts

Drug classes were defined by American Hospital Formulary Service (AHFS) codes. For each drug class of interest, we identified individuals with at least one prescription claim for a drug in the class and who had at least 11 months of MAPD enrollment from January 1, 2019 through December 31, 2019 and at least 1 month of enrollment in 2020. We further restricted our analysis to counties in Connecticut, New Jersey, and New York to ensure more consistent COVID-19 exposure in the cohort. The outcome of interest was a claim for hospitalization with a positive COVID-19 test from January 1 to June 26, 2020.

We labeled drug users based on pharmacy claims matching any generic name for the drug of interest. Drug-exposed patients were defined as those who had a prescription supply for at least 80% of the study window, starting from their first drug use date. Non-drug-exposed individuals were defined as those with no prescription for any drug in the same therapeutic class as the drug of interest in the study window. We also included one negative control associated with a known

COVID-19 confounder, glucose meters, to assess our analysis pipeline's global confounding control. We considered individuals to be exposed when they have one prescription for a glucose meter between July 1, 2019 and January 31, 2020.

### Study covariates

For each drug of interest, we extracted the following list of covariates for both drug-exposed individuals and non-drug-exposed individuals: age; gender; race; socioeconomic status index based on ZIP code; 2019 diagnoses based on three-digit ICD10 codes; pre-existing diagnoses in 2019; co-used prescriptions for the top 20 therapeutic classes; prior hospitalizations; number of primary care provider visits in 2019; count of unique drug prescriptions; routine screening adherence in 2019; flu vaccination in 2019; and special need plan.

### Controlled study without matching

We identified features using a least absolute shrinkage and selection operator (LASSO) model, including state, race, normalized SES index, normalized age, sex, primary treatment-related diagnosis, comorbidity index, occurrence of diagnosis codes, and co-use of drug therapeutic classes. Features with less than 1% prevalence in the cohort were excluded. We then fit a Cox proportional hazards model to estimate the adjusted hazard ratio for the treatment group in comparison to the control group.

### Propensity-score-matched study

For each drug of interest, we performed 1:1 propensity score matching (PSM) with a caliper of $0.25 * SD_{PS}$, where $SD_{PS}$ is the standard deviation of the propensity scores, to create matched pairs of drug-exposed and non-drug-exposed patients. The propensity score was constructed using logistic regression with the same set of features as above, and in all cases the standardized mean difference between groups was less than 0.1. As previously, we fit a Cox proportional hazards model for the drug-exposed and non-drug-exposed groups to estimate the adjusted hazard ratio.

### Supporting information

**S1 Table. List describing the clinical data analysis on clinical cohort characteristics.**
(PDF)

**S2 Table. List describing the clinical data analysis on drugs most strongly associated with COVID-19 hospitalization in the database-wide screen.**
(PDF)

**S3 Table. List of all cDNAs used and commercial vendor information.**
(PDF)

**S4 Table. List of guideRNAs used for validation.**
(XLSX)

**S1 Data. Full library guideRNA targets and sequences for the human membrane CRISPR activation library.**
(CSV)

**S2 Data. Raw scores from all genome-wide membrane screening performed in the study.**
(XLSX)

**S3 Data. Complete output table from whole transcriptomics analysis using DESeq2 pipeline (detailed in Materials and Methods).** The table includes gene symbol, baseMean, log2FoldChange, lfcSE, stat, pvalue, padj. The padj, adjusted

p-values, with FDR-correction using a cut-off (alpha) of 0.1 (the default value), were used as the statistical basis for differential expression analysis in all experiments.
(CSV)

**S1 Fig. Development of a pseudoviral based platform to screen for novel SARS-CoV-2 entry factors.** a, Schematics showing the design of vector systems used in CRISPRa screening. b, WT and ACE2 OE lines stained with or without RBD-Biotin and Streptavidin-Alexa488. c, WT and ACE2 OE cells either mock infected or infected with SARS-CoV-2 D614G Spike pseudotyped lentiviruses. d,Transfection of 293FT WT and ACE2 OE lines expressing dCas9-VP64 with pXPR_502 vector containing a CD55 targeting gRNA.
(TIFF)

**S2 Fig. Membrane-wide CRISPRa plasmid library sequencing analysis.** a, Statistics for sequencing of membrane-wide CRISPRa plasmid library. b, Distribution of gRNAs and representation in the membrane-wide CRISPRa plasmid library.
(TIFF)

**S3 Fig. Identification of the established membrane entry factors for SARS-CoV-2 in our screens and previous loss-of-function screens.** Bar plot showing the number of established membrane entry factors in the top 10% screen hits.
(TIFF)

**S4 Fig. Tissue expression analysis of top-ranking genes promoting viral entry.** Heatmap showing the overall human tissue expression patterns of top-ranking genes using GTEX v8 dataset.
(TIFF)

**S5 Fig. Functional enrichment analysis of top-ranking genes promoting viral entry.** a-b, Gene set overlap analysis using (a) gene ontology (GO) (b) pathways on top 10% of hits from each screen condition. The top GO terms or pathways of each screen condition were selected for visualization.
(TIFF)

**S6 Fig. Validation of the top-ranking genes with different variants of SARS-CoV-2 Spike pseudotyped lentivirus.** (a-b) (a) HEK293FT WT lines and (b) ACE2 OE HEK293FT lines stably overexpressing cDNAs of putative stably overexpressing cDNAs of putative SARS-CoV-2 entry factors were transduced with lentiviruses pseudotyped with either VSVG, Spike D614G variant, Spike B1.617.2 (Delta), Spike B.1.1.529 (Omicron) variant.
(TIFF)

**S7 Fig. Detection of cDNA expression in overexpression cell lines.** (a) qPCR assay of cDNA overexpressing cell lines. LogFC was calculated relative to BFP-overexpressing negative control cell lines. (b-c) Western Blots of lysates from HEK293FT WT and HEK293FT ACE2 OE cells overexpressing either BFP or KCNA6 probed with either an (b) anti-V5 or (c) anti-KCNA6 antibody. * Denotes the correct band for KCNA6 based on protein size markers.
(TIFF)

**S8 Fig. Comparison of LGMN activity and expression with different expression vectors.** a, Design of pLX304 and EF1a-2A vectors. LTR: Long Terminal Repeat, PGK: human phosphogylcerate kinase promoter, BSD: blasticidin deaminase, CMV: cytomegalovirus promoter, GOI: gene of interest, WPRE: woodchuck hepatitis virus postregulatory element, P2A: porcine teschovirus-1 2A peptide. b, Replication competent SARS-CoV-2 infection of 293FT ACE2 OE cells expressing BFP or LGMN cloned into pLX304 or EF1a-2A. Two independent experiments, six technical replicates per experiment.

Statistical analyses: two-way ANOVA with correction for multiple comparisons during hypothesis testing. *, $p < 0.05$; **, $p < 0.01$; ***, $p < 0.001$; ****, $p < 0.0001$.
(TIFF)

**S9 Fig. Replication-competent SARS-CoV-2 infection of knockout cell lines.** SARS-CoV-2 live virus infection of NCC-Stc-K140 (a) SW156 (b) and Loucy (c) cells perturbed with CRISPR-based loss-of-function constructs. Three guide-eRNAs were used per gene and results are reported for individual guide RNAs and pooled groups for analysis. Infections was repeated twice to collect replicates for all cell lines. All data represent mean with SEM. Statistical analyses: one-way ANOVA with non-target (NT) pooled as the control condition with correction for multiple comparisons during hypothesis testing. Two independent experiments, 3–6 technical replicates per experiment. *, $p < 0.05$; **, $p < 0.01$; ***, $p < 0.001$; ****, $p < 0.0001$.
(TIFF)

**S10 Fig. Confocal microscopy imaging of SARS-CoV-2 psuedovirus infection.** a-b, 293FT WT (a) or ACE2 OE (b) cells with or without exogenous LGMN expression were transduced with Spike-V5tag pseudotyped lentivirus, then stained by immunofluorescence assay imaged with a confocal microscope at 4 hr post-transduction. Blue: DAPI; green: LGMN protein; red: Spike-V5 protein. Scale Bar: 10 µm.
(TIFF)

**S11 Fig. Imaging of LGMN signal peptide mutant localization.** 293FT ACE2 OE cells transduced with LGMN or LGMN signal peptide variants were stained by immunofluorescence assay and imaged with a confocal microscope. Blue: DAPI, green: LGMN protein. Scale Bar: 10 µm.
(TIFF)

**S12 Fig. Additional single cell analyses on KCNA6 expression in the human olfactory epithelium.** a, KCNA6 genome annotations in the GRCh38 (hg38) and GRCh37 (hg19) references using NCBI Genome Data Viewer. (b) Expression of ACE2 and KCNA6 in the single-cell RNA-seq data of olfactory neuroepithelium using different versions of genome references (from Durante et al.) [51]. Cell Ranger 6.0 was used for all alignments and the expression was calculated by averaging the ACE2/KCNA6 expression in all cells and normalized to the ACE2 expression from the standard GRCh38 reference genome. (c) Average expression of ACE2/KCNA6 in olfactory epithelium using the Salmon - Alevin pipeline, calculated similarly as in panel C. The standard GRCh37/GRCh38 genome references were used. (d) UMAP depicting the olfactory epithelial cell types from two patients. The cell cluster identities were based on Durante et al. [51] (e) UMAPs depicting the expression levels of KCNA6 in individual patients. (f) Focused UMAPs of the neuronal populations showing co-expression of KCNA6 and OLIG2, a reported marker for virus-infected neuronal cells in COVID19 patient olfactory neuroepithelium.
(TIFF)

**S13 Fig. KCNA6 and LGMN overexpression doesn't alter expression of known SARS-CoV-2 host factors.** a-b, Quantitative RT-PCR analysis of ACE2 and TMPRSS2 levels upon the over-expression of indicated cDNAs in (a) WT and (b) ACE2 OE 293FT cells. c-d, Volcano plot of differentially expressed genes in BFP vs (c) KCNA6 and (d) LGMN overexpression cells across WT and ACE2 OE conditions.
(TIFF)

**S14 Fig. A drug interaction network derived from CRISPRa screen hits.** Overview of the drug-target interaction network, showing an induced subgraph of the 50 highest ranked compounds (drugs in blue; screen hits in green; potassium channel genes outlined in red).
(TIFF)

**S15 Fig.  Network analysis identifies drugs that target top screen hits, and retrospective analysis of patient claims provides real world evidence for enriched drug categories.** a, Top drug classes enriched in hits from the interaction network model by NDC and degree ratio with respect to screen hits. Asterisks indicate drug classes with at least one member targeting a potassium channel. b, Controlled study design for COVID-19 hospitalization from pharmaceutical claims data. c, Association between COVID-19 hospitalization in the unmatched study and drugs ranked highly in the drug-target interaction network. d, Real world evidence for associations between ion-channel-targeting drug classes identified in the screen and increased risk of COVID-19 hospitalization in propensity-score-matched subjects. (TIFF)

## Acknowledgments

We are grateful to members of the laboratories of Dr. Le Cong and Dr. Michael Cleary. We are grateful to Yinglong Guo, Taryn Hall, Luke Tso, Elif T. Erdemic, Kae Tanudtanud, Sheng Ren, and Kathy Tzy-Hwa Tzeng at UnitedHealth Group for the real world evidence analysis. We are grateful to Dr. Alberto Siddu, Dr. Thomas Sudhof, and Dr. Julien Sage for helpful discussion and support on understanding screen results and validation of host factors; to Dr. Joseph Wu, Dr. Mingqiang Wang, Dr. Masataka Nishiga, and Dr. Sergiu Pasca for discussion on single-cell gene expression analysis.

## Author contributions

**Conceptualization:** Ravi K. Dinesh, Chengkun Wang, Arjun Rustagi, Henry Cousins, Aimee Beck, Mengdi Wang, Le Cong.

**Formal analysis:** Trisha R. Barnard, William A Johnson.

**Funding acquisition:** Le Cong.

**Investigation:** Ravi K. Dinesh, Chengkun Wang, Yuanhao Qu, Arjun Rustagi, Henry Cousins, James Zengel, Xiaotong Wang, Trisha R. Barnard, Guangxue Xu, Tianyi Zhang, Nicholas Magazine, Aimee Beck, Lucas Miecho Heilbroner, Grace Peters-Schulze, Aaron J. Wilk, Mengdi Wang, Weishan Huang, Brooke E. Howitt, Jan Carette, Russ Altman, Catherine A. Blish, Le Cong.

**Methodology:** Trisha R. Barnard.

**Writing – original draft:** Ravi K. Dinesh.

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
