## [Decision Letter · Decision Letter 0]

PPATHOGENS-D-24-01964Membrane-wide screening identifies potential tissue-specific determinants of SARS-CoV-2 tropismPLOS Pathogens Dear Dr. Le Cong, Thank you for submitting your manuscript to PLOS Pathogens. After careful consideration, we feel that it has merit but does not fully meet PLOS Pathogens's publication criteria as it currently stands. Therefore, we invite you to submit a revised version of the manuscript that addresses the points raised during the review process. Please submit your revised manuscript within 60 days Dec 24 2024 11:59PM. If you will need more time than this to complete your revisions, please reply to this message or contact the journal office at plospathogens@plos.org. Please include the following items when submitting your revised manuscript:* A rebuttal letter that responds to each point raised by the editor and reviewer(s). You should upload this letter as a separate file labeled 'Response to Reviewers '. This file does not need to include responses to any formatting updates and technical items listed in the 'Journal Requirements' section below.* A marked-up copy of your manuscript that highlights changes made to the original version. You should upload this as a separate file labeled 'Revised Manuscript with Track Changes '.* An unmarked version of your revised paper without tracked changes. You should upload this as a separate file labeled 'Manuscript '. If you would like to make changes to your financial disclosure, competing interests statement, or data availability statement, please make these updates within the submission form at the time of resubmission. Guidelines for resubmitting your figure files are available below the reviewer comments at the end of this letter. We look forward to receiving your revised manuscript. Kind regards, Bart L. HaagmansAcademic EditorPLOS Pathogens Alexander GorbalenyaSection EditorPLOS Pathogens Michael Malim

Editor-in-Chief

PLOS Pathogens

orcid.org/0000-0002-7699-2064   **Journal Requirements:** **Additional Editor Comments (if provided):****Reviewers' Comments:** Reviewer's Responses to Questions

**Part I - Summary**

Reviewer #1: In the work by Ravi K. Dinesh et al., membrane-wide CRISPR activation screen was used to identify novel SARS-CoV-2 host factors. Validation experiments with replication-competent SARS-CoV-2 confirmed the role of newly identified host factors, including endo-lysosomal protease legumain (LGMN) and the potassium channel KCNA6. Using clinical data, they find possible associations between expression of either LGMN or KCNA6 and SARS-CoV-2 infection in human tissues. Overall, the experimental results were original and informative. The concerns/suggestions are listed below:

Major:

1. In the introduction (Pg3, Line 30), the authors reviewed current therapeutics for SARS-CoV-2, including antibodies and small molecules. There are also investigational peptidyl therapeutics targeting the fusion process (for example, Xia et al., 2021, Signal Transduct Target Ther). These therapeutics should be reviewed in a more comprehensive approach.

2. Fig 2. According to the screen result in Fig 1g and pseudovirus result in Fig 2b, LGMN OE could promote virus infection in HEK293FT ACE2 OE cells. The results in Fig 2f showed that LGMN OE could not promote live virus infection. The author should provide an explanation to this, either in the results or discussion, as well as the differential effects of LGMN in WT and ACE2OE cell line (Fig 2e and 2f)

3. Fig 3. The authors’ conclusions on Fig. 3 are: 1) The function of LGMN is dependent on ACE2; 2) the function of LGMN is the same to TMPRSS2. However, none of the above conclusion/hypothesis is fully supported by the presented data. First, comparing the data in Fig.2e and 2f, one would find that the effects of LGMN on viral infection is more profound in the absence of ACE2 OE. This seemed to be opposite to what the authors claimed. To illustrate the dependency of the function of LGMN on ACE2, double knockout cell lines should be made and examined for viral infection (as shown in subsequent section). Second, in order to claim LGMN can replace the function of TMPRSS2, the authors would need to demonstrate that LGMN and TMPRSS2 can process/cleave spike protein in the same form at the same stage of viral infection. Given the subcellular localization of LGMN as the authors stated (in endosome and lysosome), LGMN is unlikely to cleave spike protein at the very early stage of infection to trigger the membrane fusion. Rather, LGMN is more likely to function after viral internalization to promote viral uncoating or entry.

Therefore, it is suggested that the authors draw conclusions more specifically to the presented data. In addition, it is recommended to show the structural/domain organization of LGMN so that the biological relevance of the point mutations can be better understood.

4. Fig 5. Is there any specific reason for the authors to switch the cell line between 293T and 293FT in this figure?

Minor:

1. It is recommended that the quality of the constructed CRISPR libraries are described.

2. P6 line35, the description was Fig 3d instead of Fig 3e.

3. P6 line41, the description was Fig 3e instead of Fig 3d.

Reviewer #2: In this manuscript, Dinesh et al. report multiple CRISPR activation screens to identify genes that promote SARS-CoV-2 infection. Although several other groups have already published such screens, publishing additional CRISPR screen results may still have added value, as the outcomes of screens performed in different labs are quite variable. After performing preliminary validation experiments on the top-ranked genes, the authors focus on the two genes LGMN and KCNA6. Although it is not clear why these two factors were chosen for further studies, the authors assess the effect of LGMN/KCNA6 overexpression, knockout, and pharmaceutical inhibition on SARS-CoV-2 infection and the results of these validation experiments are sufficiently convincing. It appears that no other publications describe extensive validation of these host factors. Finally, the authors check whether LGMN/KCNA6 overexpression influences the expression levels of other known SARS-CoV-2 host factors and evaluate the expression of the two genes in disease-relevant tissues. Although some aspects of the manuscript are somewhat unclear/confusing, the main conclusions are relevant, sufficiently novel and are well supported by the data. However, the authors must address the following major issues before the study is suitable for publication in PLoS Pathogens:

Reviewer #3: In the manuscript by Dinesh et al, the authors performed a membrane-wide CRISPRa screen to identify critical host factors for SARS-CoV-2 entry. The findings in this manuscript are informative and provide new insights into host-directed therapeutic targets for SARS-CoV-2. However, while the presented study is informative, the authors need to provide more mechanistic insights into how LGMN or KCNA6 promote SARS-CoV-2 entry.

**Part II – Major Issues: Key Experiments Required for Acceptance**

Reviewer #1: (No Response)

Reviewer #2: • The authors performed a validation experiment with a subpool CRISPRa library targeting top-ranked genes from their initial screen. However, discussion about the design and outcomes of this experiment are less than the bare minimum. Since this is a validation screen, I would expect a statement (and data) about how many of the 523 and 542 genes could be validated. Instead, the authors only mention that their top-ranked sgRNAs perform consistently, irrespective of the use of WT or D614G Spike. The authors should present and discuss the outcomes of this validation screen more clearly or, in case their results were disappointing, consider removing this experiment from the paper altogether, as its value appears to be very limited.

• Figure 4d: In this experiment, the authors used three individual sgRNAs per gene to generate three separate knockout cell lines that are infected with virus. This strategy is common practice in CRISPR screen validation because testing multiple sgRNAs reduces the chance that the observed effects are due to off-target editing. Demonstrating that three unique sgRNAs all reduce virus infection would convincingly show that this is really due to on-target knockout. However, the authors aggregated data of the different guides for each gene and present only the mean of the guides. Data for each sgRNA should be presented separately as this is one of the most important figures in the paper and it should be as convincing as possible.

• Page 6, lines 1-2: the authors conclude that CD7, KCNA6, and LGMN overexpression promote infection, but an effect of CD7 overexpression is not shown in Figures 2c and d. This should be corrected. Also, the authors may consider to adjust the phrase “similarly promote infection” as it suggests that effect sizes are the same for the different genes.

• Figure 5a: The authors conclude from this figure that the stimulating effect of LGMN/KCNA6 overexpression on infection is dependent on ACE2. However, it is not clear whether the ACE2-KO cells are still infected by the virus. If ACE2-KO cannot be infected, it is quite unsurprising that overexpression of other genes will not have an effect but this would not prove that their functions depend on ACE2. Therefore, the authors should either show that the cells are infected, or remove the conclusion about dependency of LGMN/KCNA6 on ACE2.

• Figure 5d: The authors performed transcriptome analysis on LGMN/KCNA6 overexpressing cells but it is not clear to me what exactly is shown in the figure. The experiment is hardly explained in the text, it is not clear whether labels at the bottom represent knockout or overexpression, and the color scale is not labelled. This needs to be clarified.

Reviewer #3: 1.The authors have validated several candidates by cDNA overexpression and observed minor phenotype following pseudovirus and authentic SARS-CoV-2 infection. However, most of the top hits were not validated. The rationality for choosing the candidates for validation is not explained in the manuscript. Notably, LGMN or KCNA6 is not even the positive hits in 293T-ACE2 cells screens. (KCNA rank in different condition screens (#68, #309�#3960,#3350).

2.The authors compared several LOF screen results with their GOF screen findings. However, the authors haven’t done this comparasion with previous GOF screens (such as PMID: 35879413, 35879412 etc). Is there any hits overlap between this study and previous GOF screens?

3.The cell lines used for SARS-CoV-2 infection is not well characterized cell lines for CoVs infection. Do these hits identified in this study have similar phenotype in cells most used for CoV infection such as in Vero, Calu-3?

4.The authors didn’t provide more molecular mechanisms by which LGMN or KCNA6 promote viral entry. Does protease LGMN promote viral entry by cleaving and activating Spike protein like TMPRSS2?

**Part III – Minor Issues: Editorial and Data Presentation Modifications**

Reviewer #1: (No Response)

Reviewer #2: • Page 3, lines 35-36: This sentence suggests that inhibitors of viral genome replication are in general less effective that entry inhibitors, but many effective replication inhibitors exist for other viruses. The authors should consider weakening this statement.

• Page 4, lines 18-19: “We then confirmed that these cells were capable of sgRNA-directed gene activation”. It is not clear how it was shown that sgRNA-directed gene activation is working, as Figure S1c appears to only show that the cells can be infected.

• Page 4, line 39: It is specifically mentioned that the previously described receptor NRP1 is found in the screens, with a reference to Figures 1d-e, g. However, this gene is not shown in the figures.

• References to Figures 3d and 3e appear to be swapped.

Reviewer #3: The study would benefit from more robust virologic assays. The infection experiments utilize pseudovirus assys and luciferase as a read out. A more direct measure of viral replication or susceptibility would be of value. This would be ideally done by plaque assay.

PLOS authors have the option to publish the peer review history of their article (what does this mean? ). If published, this will include your full peer review and any attached files.

**Do you want your identity to be public for this peer review?** For information about this choice, including consent withdrawal, please see our Privacy Policy .

Reviewer #1: **Yes: ** Jia Liu

Reviewer #2: No

Reviewer #3: No

---

## [Decision Letter · Decision Letter 1]

Dear dr. Le Cong,

We are pleased to inform you that your manuscript 'Membrane-wide screening identifies potential tissue-specific determinants of SARS-CoV-2 tropism' has been provisionally accepted for publication in PLOS Pathogens.

Before your manuscript can be formally accepted you will need to complete some formatting changes, which you will receive in a follow up email. A member of our team will be in touch with a set of requests. In addition, the authors should address minor comments raised by one of the reviewers and Editors (see below) at the proof stage. 

Best regards,

Bart L. Haagmans

Academic Editor

PLOS Pathogens

Alexander Gorbalenya

Section Editor

PLOS Pathogens

Sumita Bhaduri-McIntosh

Editor-in-Chief

PLOS Pathogens

orcid.org/0000-0003-2946-9497

Michael Malim

Editor-in-Chief

PLOS Pathogens

orcid.org/0000-0002-7699-2064

Editor Comments:

Please rename SARS-CoV-1 (colloquialism) to SARS-CoV, according to the established practice.

Reviewer Comments (if any, and for reference):

Reviewer's Responses to Questions

**Part I - Summary**

Reviewer #1: The authors have addressed all my concerns.

Reviewer #2: (No Response)

Reviewer #3: The authors performed additional experiments and analysis to address my concerns, which significantly improved the the manuscript. I have no other concerns.

**Part II – Major Issues: Key Experiments Required for Acceptance**

Reviewer #1: The authors have addressed all my concerns.

Reviewer #2: (No Response)

Reviewer #3: (No Response)

**Part III – Minor Issues: Editorial and Data Presentation Modifications**

Reviewer #1: (No Response)

Reviewer #2: • The authors included new imaging data showing the localization of LGMN and Spike. Based on these data, they conclude that Spike colocalizes with LGMN and that this colocalization is more pronounced in the context of ACE2 overexpression. I think that these statements about colocalization should be removed, because proving colocalization requires more than showing that two proteins are present within the same cell.

• Related to this, I suggest to crop the images in the new figure S11 to make the relevant cell more easily visible. It is currently very difficult to see the differences between signal peptide mutants.

• The authors have shown infection data for each individual sgRNA in Supplementary figure 9, as I previously requested. It is important to clearly indicate in the manuscript that Supplementary figure 9 shows the exact same data as the corresponding main figures (Fig. 3 c&d and Fig. 4d?), since it is not acceptable to show the same data twice without mentioning this. I would suggest referring to Fig. S9 in each legend of the corresponding main figures and vice versa, to make this absolutely clear.

Apart from these minor issues, the authors have adequately addressed my original concerns.

Reviewer #3: (No Response)

PLOS authors have the option to publish the peer review history of their article (what does this mean? ). If published, this will include your full peer review and any attached files.

**Do you want your identity to be public for this peer review?** For information about this choice, including consent withdrawal, please see our Privacy Policy .

Reviewer #1: **Yes: ** Jia Liu

Reviewer #2: No

Reviewer #3: No

---

## [Editor Report · Acceptance letter]

Dear Dr Cong,

We are delighted to inform you that your manuscript, "Membrane-wide screening identifies potential tissue-specific determinants of SARS-CoV-2 tropism," has been formally accepted for publication in PLOS Pathogens.

Best regards,

Sumita Bhaduri-McIntosh

Editor-in-Chief

PLOS Pathogens

orcid.org/0000-0003-2946-9497

Michael Malim

Editor-in-Chief

PLOS Pathogens

orcid.org/0000-0002-7699-2064